# Y chromosome toxicity does not contribute to sex-specific differences in longevity

Rénald Delanoue [1,2] ✉, Charlène Clot [1,2], Chloé Leray[1], Thomas Pihl [1] & Bruno Hudry [1] ✉

While sex chromosomes carry sex-determining genes, they also often differ from autosomes in size and composition, consisting mainly of silenced heterochromatic repetitive DNA. Even though Y chromosomes show structural heteromorphism, the functional significance of such differences remains elusive. Correlative studies suggest that the amount of Y chromosome heterochromatin might be responsible for several male-specific traits, including sex-specific differences in longevity observed across a wide spectrum of species, including humans. However, experimental models to test this hypothesis have been lacking. Here we use the *Drosophila melanogaster* Y chromosome to investigate the relevance of sex chromosome heterochromatin in somatic organs in vivo. Using CRISPR–Cas9, we generated a library of Y chromosomes with variable levels of heterochromatin. We show that these different Y chromosomes can disrupt gene silencing in *trans*, on other chromosomes, by sequestering core components of the heterochromatin machinery. This effect is positively correlated to the level of Y heterochromatin. However, we also find that the ability of the Y chromosome to affect genome-wide heterochromatin does not generate physiological sex differences, including sexual dimorphism in longevity. Instead, we discovered that it is the phenotypic sex, female or male, that controls sex-specific differences in lifespan, rather than the presence of a Y chromosome. Altogether, our findings dismiss the 'toxic Y' hypothesis that postulates that the Y chromosome leads to reduced lifespan in XY individuals.

Many species show sex-specific differences in lifespan[1,2]. A popular model to explain this finding is that the sex chromosomes contribute to ageing through a 'toxic Y' effect[3]. This suggests that the accumulation of repetitive DNA on the Y chromosome results in the death of males at a younger age. Several recent meta-analyses, using data from hundreds of species from the four major clades of tetrapods (amphibians, reptiles, birds and mammals), discovered that the homogametic sex, on average, lives 17.6% longer than the heterogametic sex[4–7]. For example, XX female lions, Asian elephants and killer whales live 70% longer than their XY male counterparts[6,7]. In humans, XX individuals live typically 7.8% longer than XY individuals[8]. Compared to a control population, individuals with an XXY karyotype (Klinefelter syndrome) have a 2-year reduction in longevity and those with an XYY karyotype have a 10-year reduction[9]. These data suggest a Y effect in humans too. Interestingly, certain animal classes show an opposite effect. In some birds and reptiles, males are the homogametic sex (ZZ) while females are ZW, with W being equivalent to the Y chromosome. In the common toad and the American magpie, ZZ males outlive females[4,6,7]; their longevity is 70% greater. This observation extends to another kingdom of life. In dioecious plants with heteromorphic sex chromosomes, XY individuals experience reduced lifespan[10].

[1]Institut de Biologie Valrose, Université Côte d'Azur, CNRS, INSERM, Nice, France. [2]These authors contributed equally: Rénald Delanoue, Charlène Clot. ✉e-mail: renald.delanoue@univ-cotedazur.fr; bruno.hudry@univ-cotedazur.fr

How can the toxicity of the Y chromosome be explained at the molecular level? Experiments in *Drosophila melanogaster* have shown that the Y chromosome decreases genome-wide enrichment patterns of the repressive chromatin modifications, tri- or di-methylated histone H3 Lys9 (H3K9me3/2)[11–14]. The Y chromosome, composed mainly of silenced repetitive DNA, acts in *trans* by sequestering core components of the heterochromatin machinery. These negative impacts of the Y chromosome on constitutive heterochromatin increase with age and silenced repeated sequences can become derepressed specifically in old males[12,13,15]. This heterochromatin loss and age-associated increase of repeat expression are postulated to be the molecular drivers underlying Y toxicity and reduced survival of XY individuals[3]. In this model, the detrimental impacts of the Y chromosome are predicted to be determined only by its size and its capacity to deplete heterochromatin factors from the rest of the genome[3,11–13].

Direct functional tests of the toxic Y theory are currently lacking, due to the absence of suitable genetic models. Here, using CRISPR–Cas9 in flies, we generated a library of Y chromosomes of different sizes. The *Drosophila* Y chromosome contains only a few protein-coding genes expressed exclusively in the male germline and is entirely heterochromatic in somatic cells[16,17]. As a result, our fly strains only carry the epigenetic effect of the Y chromosome in somatic cells. Our study demonstrates that the presence, number, or size of the Y chromosome do not effect sexual dimorphism in longevity. Our findings strongly refute the suggestion that the Y chromosome has a toxic effect and shortens the male lifespan.

## Generation of Y chromosomes of different sizes using CRISPR

Sex-specific differences in lifespan prompted us to investigate a proposed molecular underpinning: the size-dependent detrimental effect of Y chromosome heterochromatin[3]. We developed an original technique in the fly to engineer large-scale deletions using CRISPR–Cas9 and create isogenic animals with polymorphic Y chromosomes of different lengths. To test whether chromosomal truncations could be recovered with CRISPR–Cas9, we designed sets of single guide RNAs (sgRNAs) that target four specific Y loci: *Suppressor of Stellate* (*Su(Ste)*), *flagrante delicto Y* (*FDY*), *Protein phosphatase 1 Y-linked 2* (*Pp1-Y2*) and the Y centromere (Fig. 1a). We selected loci containing Y-specific repeats where sgRNAs are predicted to generate multiple cleavages specific to the Y chromosome. For example, on the basis of the Y chromosome DNA sequence, our sgRNAs targeting the *Su(Ste)* locus are predicted to cut 223 times (Fig. 1a). We suggested that the targeted chromosome arm will be cleaved beyond the limits of cell repair and will be selectively eliminated at cell division (Fig. 1b).

Somatic expression of these sgRNAs in wing and eye imaginal discs induced DNA double-strand breaks specifically in XY male cells and not in XX female cells (Fig. 1c and Extended Data Fig. 1a). Importantly, we also observed the loss of a Y-linked phenotypic marker located near the telomere of the long arm: *Bar* (Fig. 1d and Extended Data Fig. 1b). The *Bar* marker reduced fly eyes to rectangular vertical bars. This marker was lost when flies were treated with the three sgRNAs targeting the long arm of the Y chromosome in somatic cells of the eye disc. This marker was not lost in cells treated with an sgRNA that target the short arm (near the *Pp1-Y2* locus, Fig. 1d). The effectiveness of marker loss was proportional to the predicted number of cuts: using sgRNAs that cut the Y chromosome 223 times induced loss of the long arm marker in >50% of the eyes (Fig. 1d). The above results suggested that cutting a chromosome many times at a specific location frequently leads to the loss of the fragment without a centromere, as reported in mouse and human embryonic stem cells[18–20].

To obtain whole animals with different Y chromosomes, we expressed these sgRNAs in the male germline. Up to 42% of the $F_1$ male progeny appeared to have modified Y chromosomes, as indicated by the loss of several Y markers located along the chromosome (Extended Data Fig. 1b,c). We assessed if these chromosomes carried translocations, internal deletions or terminal deficiencies. Using a Y-linked fluorescent transgene, we counted the number of red fluorescent protein (RFP)-positive $F_1$ females, corresponding to potential translocations of the RFP marker from the Y chromosome to another chromosome. Depending on the sgRNA, translocation frequency was between zero and 0.5% (Extended Data Fig. 1d), indicating that the interchromosomal translocation frequency is at least 80 times lower than the incidence of modified Y chromosomes. Using several X-linked GFP transgenes, we established that the modified Y chromosomes did not carry additional genetic material derived from the X chromosome, corresponding to X-to-Y translocations (Extended Data Fig. 1e). Next, to determine the frequency of internal versus terminal deletions, we selected 90 modified Y chromosomes and carried out a PCR survey of all the Y chromosome coding genes. We identified 46 terminal deletions, 44 complete Y deletions (X0 males) and no internal deletions (Extended Data Fig. 1f and Extended Data Table 1). Altogether, these results indicate that our approach allows the generation of terminal Y chromosome deletions, at high frequency. We then probed if the sgRNAs produced fragments of the anticipated size on the basis of the location of their predicted cleavage sites along the chromosome. We used a Y chromosome with two genetic markers: one on the tip of the short arm and the other on the long arm. With the sgRNA targeting the *Pp1-Y2* locus on the short arm, we obtained truncations of the targeted arm but also at lower frequency long arm deletions (or even double short arm and long arm truncations; Extended Data Fig. 1b). Similar results were found with the sgRNAs directed against the *FDY* locus located on the long arm (Extended Data Fig. 1b). Our approach allows the production of specific Y deletions but, at low occurrences, unexpected truncations are also created. This illustrates that the Y chromosome is very repetitive and poorly characterized with <50% of its sequence known. We established a library of isogenic lines derived from single modified males and characterized five Y deletions in detail.

To characterize the Y library at the molecular level, we first mapped the Y chromosome fragments absent in the different deletions by carrying out a PCR survey of all the Y chromosome coding genes (Fig. 1e)[21,22]. We then confirmed the size of these chromosomes using flow cytometry (Extended Data Fig. 1g) and mitotic neuroblast spreads (Extended Data Fig. 1h). We found that the absolute size of the Y chromosomes recovered varies drastically from 21% to 72% of the wild-type Y (Extended Data Fig. 1g,h). We finally performed fluorescent in situ hybridization (FISH) against Y-specific satellite DNA located specifically on the short arm or the long arm of the Y chromosome (Fig. 1f and Extended Data Fig. 1i)[23,24]. The results were perfectly consistent with the PCR survey for all five deletion lines.

We then sought to functionally characterize the Y library. The *Drosophila* Y chromosome possesses only three types of functionally important genetic elements: (1) coding genes, at least six of which are essential for male fertility, (2) hundreds of copies of the transcription unit coding for the ribosomal RNAs and (3) the *Su(Ste)* repeats that produce PIWI-interacting RNAs (piRNAs) targeting the X-linked repetitive *Stellate* genes[16,17]. As expected, all the males carrying Y chromosomes from the library were sterile, as they all lacked at least one essential coding gene (Fig. 1g). The three truncations missing the Y-linked rDNA locus were unable to complement X chromosomes lacking the rDNA repeats in XXY females (Fig. 1h)[25]. All, except one ($Y_{72}$), truncation lines accumulated Stellate crystalline aggregates in spermatocytes, consistent with deletion of the Y-linked *Su(Ste)* piRNA cluster from these chromosomes (Fig. 1i)[26].

Lastly, we tested the capacity of the sgRNAs to produce, always, the same terminal truncations. All the deletions from the library were recovered at a comparable frequency in three independent experiments, indicating the efficacy and reproducibility of our approach (Extended Data Fig. 1j). Using CRISPR–Cas9, we generated a library of heteromorphic Y chromosomes of different sizes (Fig. 1j) that we labelled $Y_{21}$, $Y_{26}$, $Y_{53}$, $Y_{69}$ and $Y_{72}$, all derived from a unique control Y chromosome.

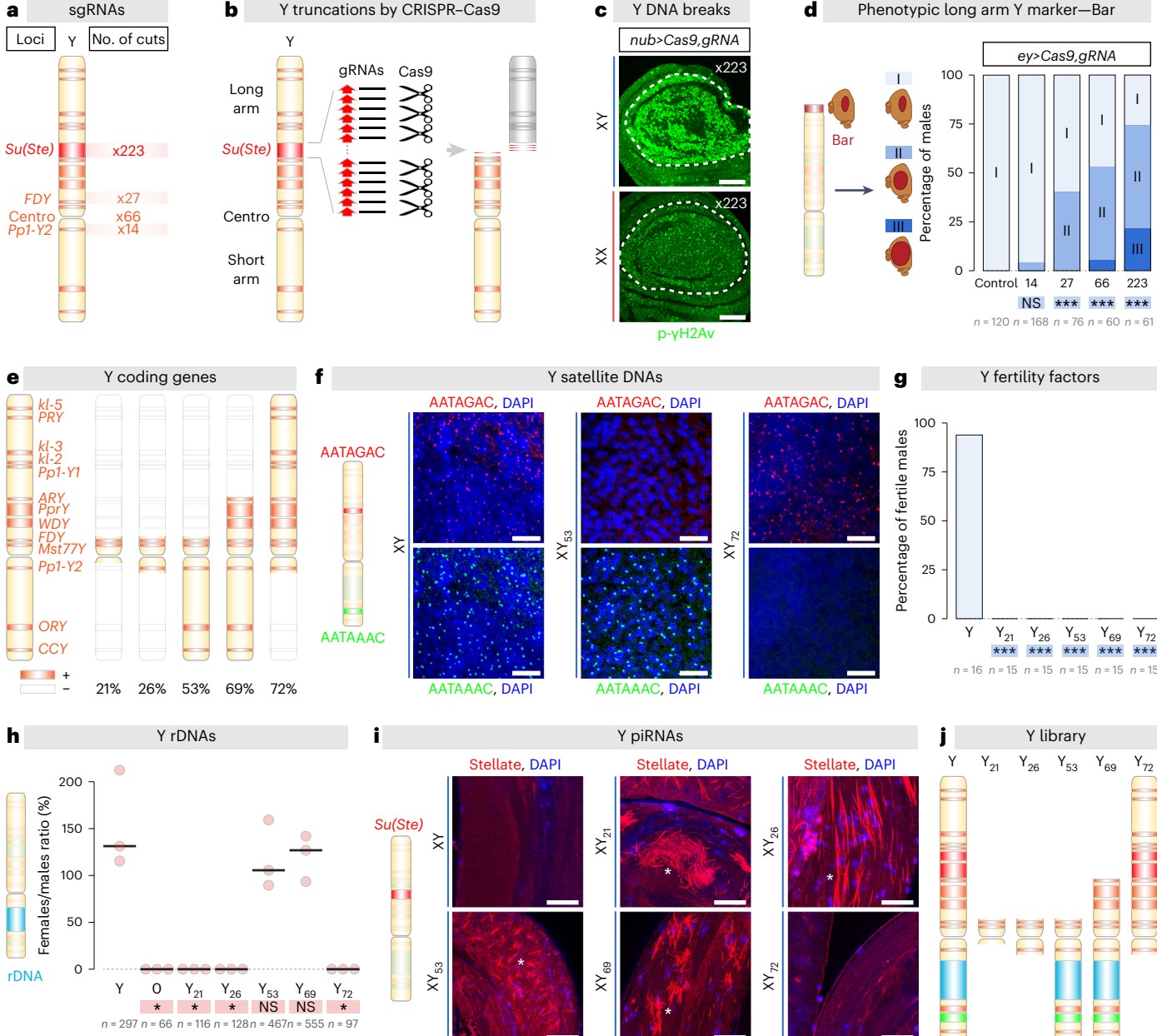

**Fig. 1 | Production of polymorphic Y chromosomes of different sizes using CRISPR. a**, Diagram displaying Y loci targeted by our sgRNAs. The number of cuts predicted at each locus is indicated. **b**, *Drosophila* Y chromosome diagram showing the localization of the centromere (Centro). Multiple cleavages were induced by sgRNAs (black lines) targeting one specific locus (here *Su(Ste)*), containing Y-specific repeats (red arrows), to selectively eliminate a portion (in grey) of the Y chromosome in vivo. **c**, Wing discs of third-instar expressing Cas9 and the *sgRNA x223* under the control of the *nub-Gal4* driver (expressed in the wing pouch, dashed lines) and stained for the DNA double-strand breaks marker p-γH2Av (green). Immunohistochemical analyses were repeated three independent times. Scale bars, 50 μm. **d**, Percentage of eyes exhibiting different phenotypes (Y-linked Bar (I), intermediate (II) or wild type (III)) in males expressing Cas9 and an sgRNA under the control of the *ey-Gal4* driver (expressed in eye disc). *P* values from one-sided ANOVA are non-significant (NS) *P* = 0.11, ***P* < 0.0001. **e**, PCR genotyping analyses of the Y-linked coding genes. Present

and deleted genes are symbolized by orange and white boxes, respectively. The percentage of the Y chromosome remaining is also displayed. **f**, DNA-FISH analysis of larval brains of males carrying the indicated Y chromosome. Probe sequences targeting Y satellite DNAs are specified by the coloured text, DAPI (in blue, DNA). Scale bars, 10 μm. FISH analyses were repeated three independent times. **g**, Fertility of males carrying different Y chromosomes. *P* values from one-sided ANOVA are ***P* = 0.0005. **h**, Viability of XXY females carrying X chromosomes with deleted rRNA locus. Such females must carry a Y chromosome with an intact rRNA locus to survive. *P* values from one-sided ANOVA are (NS) *P* > 0.34, **P* = 0.0143. **i**, Testes from males carrying different Y chromosomes stained with an anti-Stellate antibody (red, Stellate aggregates (asterisks)) and DAPI (blue, DNA). Scale bars, 25 μm. Immunohistochemical analyses were repeated three independent times. **j**, Diagram of the Y chromosome truncations. *n* = number of flies. Asterisks highlighting comparisons within female and male datasets are displayed in red and blue boxes.

## The Y chromosome sequesters the heterochromatin machinery

We first explored if the different Y truncations can affect heterochromatin formation on other chromosomes, as predicted. We turned to

a well-established paradigm: the graded effect of the Y chromosome on position-effect variegation (PEV)[27]. One of the best examples of PEV is seen when the *white* (*w*) gene is translocated from euchromatin to a new position in the pericentric heterochromatin of the X chromosome:

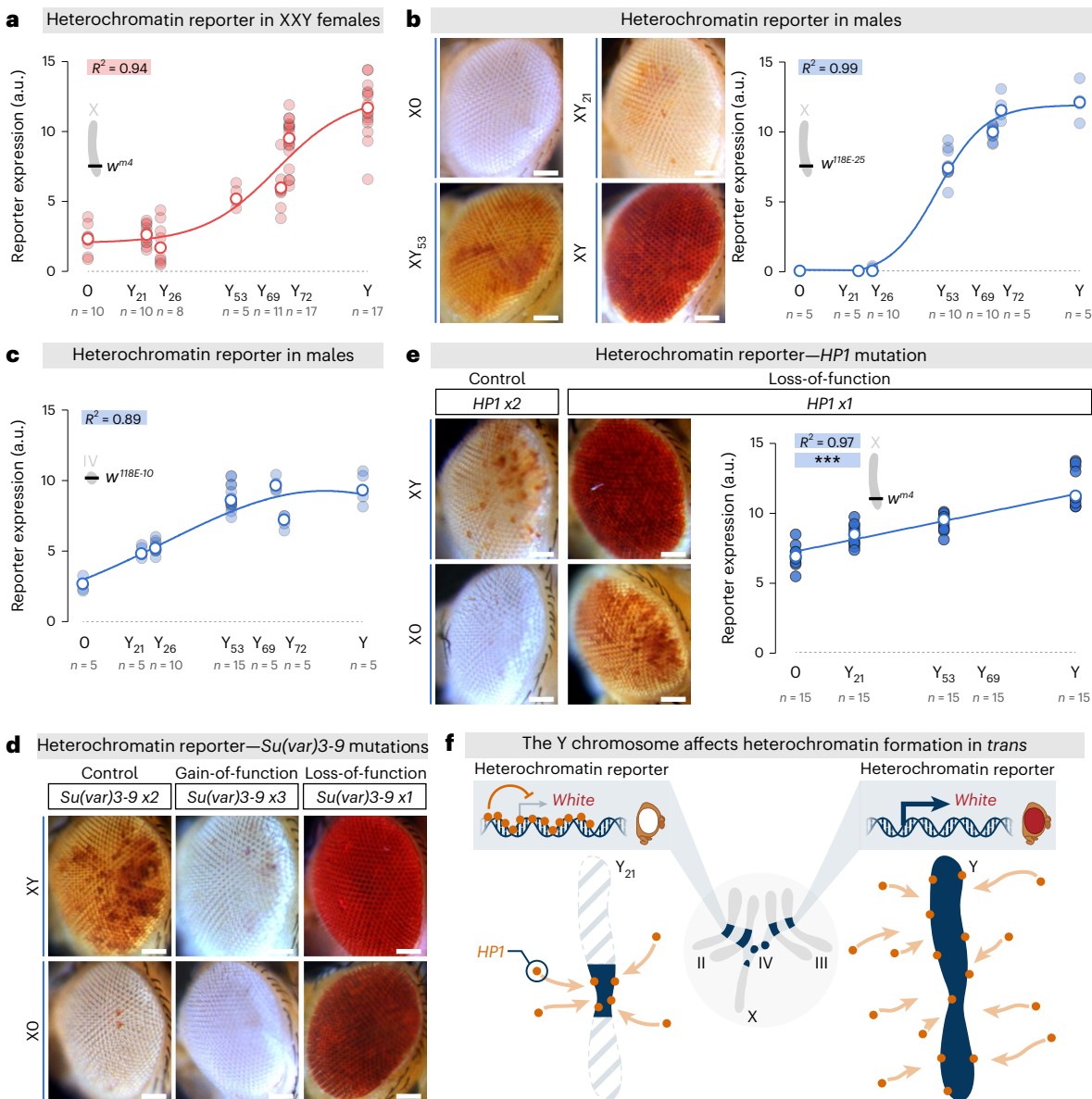

**Fig. 2 | The Y chromosome affects heterochromatin loci in *trans*, in proportion to its size. a–c**, Effects of Y chromosome size on the expression of the *white*-based heterochromatin reporters *w^m4* in XXY females (**a**), *w^118E-25* in males (**b**) and *w^118E-10* in males (**c**). Red eye-pigment levels were determined by measuring absorbance at 480 nm. $R^2$ from nonlinear regressions, using sigmoidal dose–response equations. **d**, Representative pictures comparing *w^m4* reporter expression in XY and X0 males carrying two (control), three (gain-of-function) or one (loss-of-function) copies of the *Su(var)3-9* gene. **e**, Effect of Y chromosome size on the expression of the *w^m4* reporter in males missing one copy of *HP1*. *P* value and $R^2$ from simple linear regression, \*\*\**P* < 0.0001. **f**, When the *white* (*w*) gene is artificially translocated in the constitutive heterochromatin (dark blue), the presence of a Y chromosome sequesters key factors, like HP1, allowing *w* expression and red-eye phenotype (on the right). Reduction of Y size, here by a factor of five (with Y_{21}), releases HP1 proteins and triggers *w cis*-heterochromatin inactivation giving a white-eye phenotype (on the left). In all panels, *n* = number of repeats (each repeat containing five flies). Scale bars, 50 μm. See also Extended Data Fig. 2.

a mutation named *In(1)w^m4* (ref. 28). Normally, the *w* gene is expressed in every cell of the adult *Drosophila* eye resulting in a red-eye phenotype. In this *In(1)w^m4* mutant, *w* undergoes a *cis*-heterochromatin inactivation in a random percentage of cells giving a mosaic phenotype: red-white mosaic eye colour. A strong increase in *w* expression is found in *In(1) w^m4* flies carrying an additional Y chromosome (typically engineered XXY females), whereas robust reduction is observed in males without a Y chromosome[29,30].

We found that the addition of an increasing amount of Y heterochromatin material in XX females resulted in progressive suppression of *In(1)w^m4* variegation and a complete red-eye phenotype in XXY females (Fig. 2a). To confirm these results, we also tested the impact of the Y truncations in males on the expression of two additional *white*-based heterochromatin reporters, located in the pericentric heterochromatin of the X (Fig. 2b) and the IV chromosomes (Fig. 2c)[29]. In both cases, our Y fragments decreased variegation, with effects positively correlated with Y chromosome size. For example, dividing Y size by two (with Y_{53}) reduced *w* expression by two for both reporters (Fig. 2b,c). Finally, using a PEV reporter containing a *LacZ* gene, we visualized the effects of Y truncation at a single-cell level by examining the impact of Y chromosome size on gene expression in the heterochromatin of the Malpighian tubule cells (renal-like system)[31]. Again, the data indicate that the Y chromosome disturbs constitutive heterochromatin formation in *trans* and thus impacts transcription, with reporter gene silencing being positively correlated to the size of Y heterochromatin (Extended Data Fig. 2).

What is the molecular mechanism underlying these effects of the Y chromosome? It has previously been shown that the Y chromosome can act by depleting essential chromatin machinery from the rest of the genome[16,28,32–34]. We tested if the Y chromosomes from our library could recapitulate these results. We manipulated the dose of two major genes linked to heterochromatin formation: *Suppressor of variegation 3-9* (*Su(var)3-9*) and *Heterochromatin Protein 1* (*HP1*). Su(var)3-9 is a histone methyltransferase that specifically trimethylates lysine 9 of histone H3. In *Drosophila*, this epigenetic mark has a central function in the establishment of the constitutive heterochromatin at pericentric regions and the associated gene silencing by recruiting HP1 (refs. [27,35,36]). Adding an extra copy of *Su(var)3-9* restored PEV and rescued the effect of the Y chromosome in XY males (Fig. 2d). Conversely, removing one copy of *Su(var)3-9* suppressed PEV, dominating the strong enhancer effect caused by loss of the Y chromosome in X0 males (Fig. 2d). Likewise, taking away one dose of *HP1* decreased PEV in males, leading to more cells expressing the *w* gene in the eye (Fig. 2e). The *HP1* mutation showed a co-operative effect with the size of the Y chromosome. The strength of PEV suppression was positively correlated with the length of the Y fragments present.

These genetic interactions confirmed that the truncated Y chromosomes can affect heterochromatin formation on other chromosomes in *trans* and impact gene silencing by titrating heterochromatin factors, such as HP1. The intensity of these effects in the male epigenome are positively correlated to the size of the Y chromosome (Fig. 2f).

## The Y chromosome does not reduce longevity in males

Are the epigenetic effects of the Y chromosome physiologically significant? The newly engineered polymorphic Y chromosomes allowed us to functionally test whether sex chromosome heteromorphism generates physiological sex differences. To do so, we compared XX females with our collection of males carrying a specific amount of Y heterochromatin material and examined some key male–female differences.

Somatic loss of the Y chromosome is the most common known acquired human mutation[37,38]. Its frequency increases with age and is associated with decreased survival time from all causes, including cancers[39]. To our knowledge, no causal link has been established. We first used CRISPR–Cas9-mediated targeted Y chromosome elimination and shortening to model Y chromosome loss in a cancer model. Intestinal stem cell (ISC) and adult-specific interference with *Notch* (*N*) results in tissue overgrowth, reminiscent of gastrointestinal cancers and presents sex differences in tumour incidence[40]. As a result, we started by focusing our analyses in the adult midgut. At the cellular level, fly adult ISCs exhibit sexually dimorphic proliferative behaviour[40]. The amount of Y chromosome heterochromatin did not impact this sex-specific difference in adult ISC proliferation (Fig. 3a). Female flies also exhibit a rapid proliferative response to dextran sodium sulfate (DSS)-induced damage of the intestinal epithelium[40]. This response is less pronounced in males. Y dosage did not affect DSS-induced proliferation in males (Fig. 3a). As mentioned previously, females are more prone to genetically induced tumours[40]. Indeed, adult-specific interference with *N* results in tissue overgrowth in female but not male midguts. Again, Y chromosome size and presence were dispensable for this sexual dimorphism in tumour incidence (Fig. 3b).

We then tested pupal size[41,42], adult weight[42] and starvation resistance[43]. These characteristics display differences between males and females. The molecular and cellular mechanisms driving these dimorphisms are still incompletely understood and the impact of the Y chromosome size have never been tested. We wanted also to evaluate these parameters since they have been shown to affect lifespan. The size of the Y chromosome heterochromatin material did not contribute to any of these sex-specific differences (Fig. 3c–e).

Finally, we tested if the epigenetic effect of the Y chromosome is at the origin of the unexplained sex-specific difference observed in lifespan across a wide spectrum of species[4–7,10,44]. Very surprisingly and unexpectedly, Y chromosome heterochromatin did not modulate male lifespan at all (Fig. 4a), arguing against Y-based toxicity. Indeed, reducing Y material by five (using $Y_{21}$) or by two (using $Y_{53}$) did not extend male lifespan as predicted by the toxic Y hypothesis. To confirm these results, we performed experiments with more drastic manipulations including deletion or duplication of the entire Y chromosome (Fig. 4b). Consistent with the results above, X0, XY and XYY males displayed identical longevity. To push the system even further, we removed one copy of *HP1* (Fig. 4c). It is important to note that at the molecular level, this very same *HP1* mutation showed a co-operative effect with the size of the Y chromosome on gene silencing and constitutive heterochromatin formation (Fig. 2e). Once again, heterozygous *HP1* mutant males carrying Y chromosomes of different sizes displayed the same lifespan as control males. Altogether, these data dismiss the suggestion that the Y chromosome leads to reduced longevity of XY individuals.

Ageing in *Drosophila* is characterized by loss of repressive heterochromatin and loss of silencing of reporter genes and transposable elements in constitutive heterochromatin regions[31,45–48]. We did observe an age-related loss of heterochromatin stability in intestinal enterocytes due to the loss and dispersion of the histone modifications H3K9me2 (Extended Data Fig. 3a) and H3K9me3 (Extended Data Fig. 3b). These chromatin changes were also observed in another tissue, the Malpighian tubules (Extended Data Fig. 3c). Derepression of a heterochromatin reporter (Extended Data Fig. 3d) and the *copia* retrotransposon (Extended Data Fig. 3e) accompanied this reduction in constitutive heterochromatin. Importantly, comparing XX females, XY and X0 males, we discovered that all these modifications were neither sex-specific/biased nor modified by the presence and/or the size of the Y chromosome (Extended Data Fig. 3a–e). These results confirmed that the breakdown in transposable element silencing and loss of repressive heterochromatin is not a contributing factor to sex differences in longevity. Altogether, these data definitively rule out the influence of the Y chromosome on the genome-wide chromatin landscape as the molecular underpinning of the sex-specific differences in lifespan.

What other mechanism might account for differences in longevity? As the sex gap in longevity did not follow chromosomal sex, we tested whether phenotypic sex is the determining factor. Along with heterochromatin accumulation, heteromorphic sex chromosomes are defined by the presence of sex-determining genes[49]. Although their importance is established for many sexually dimorphic anatomical features and behaviours, their contribution to disparities in longevity has been overlooked[49,50]. In *Drosophila*, the presence of two X chromosomes leads to the female-specific expression of *transformer^F* (*tra^F*), the master regulator of female identity[50]. Genetic manipulations of this sex determinant allowed us to uncouple chromosomal sex from phenotypic sex. Indeed, *tra^F* null mutant XX females are phenotypically male, while XY males expressing *tra^F* are phenotypically female. Masculinized *tra^F* knockout females have shorter lifespans (Fig. 4d and Extended Data Fig. 3f). Conversely, constitutively feminized males expressing *tra^F* displayed an extended lifespan (Fig. 4d and Extended Data Fig. 3f). Under these conditions, sex differences in longevity are completely abolished (Fig. 4d) and chromosomal and phenotypic sexes are uncoupled. These experiments indicate that phenotypic sex, rather than the presence of a Y chromosome, controls sex-specific differences in lifespan.

## Discussion

Using CRISPR–Cas9, we generated a library of sex chromosomes with different degrees of structural heteromorphism. These genetic tools allowed us to functionally test the epigenetic impact of the Y chromosome as well as the biological significance of these effects.

Collectively, our results demonstrate that Y chromosome heterochromatin does not cause or have a notable contribution to the tested physiological differences between males and females: this includes

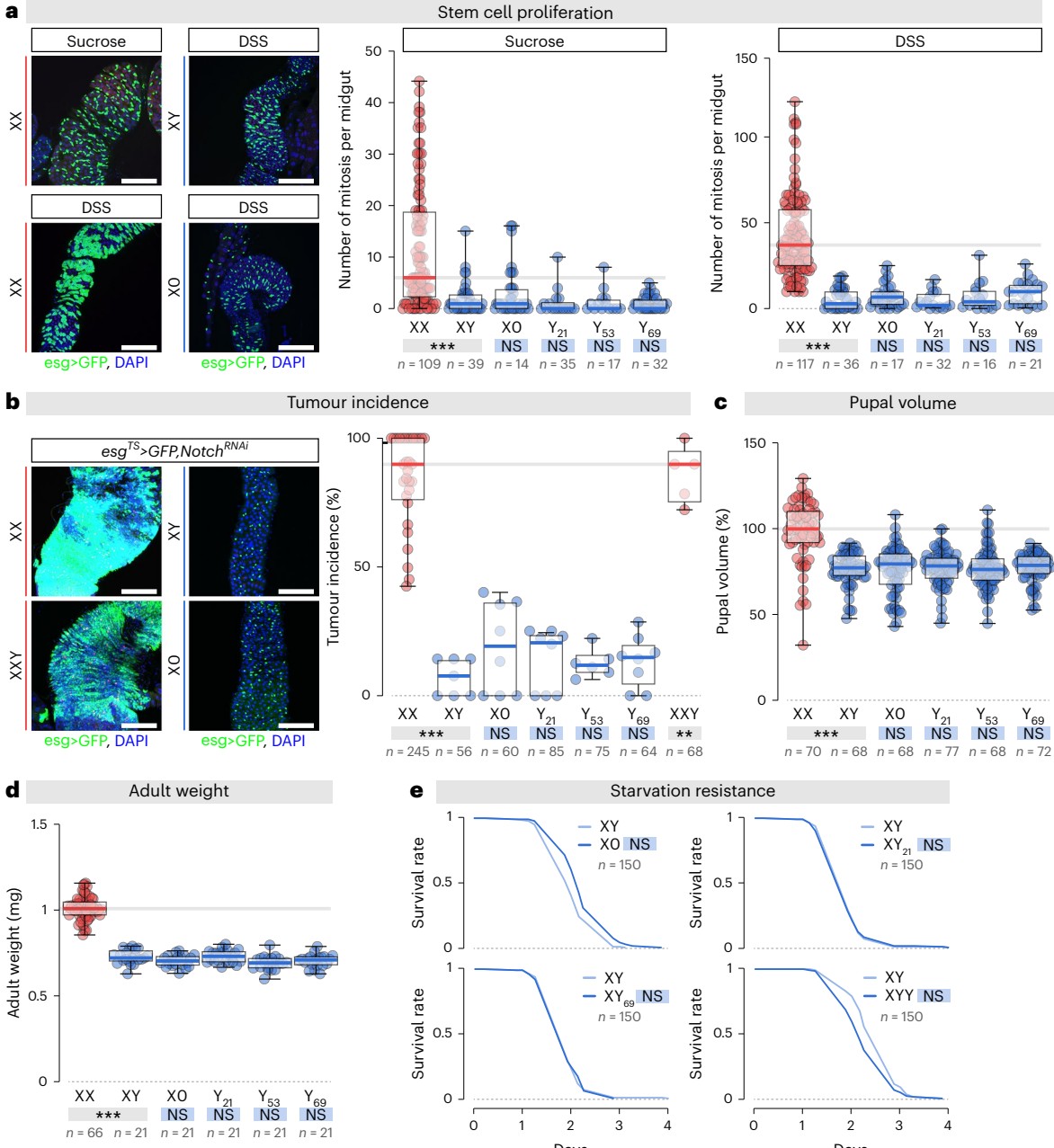

**Fig. 3 | The ability of the Y chromosome to affect heterochromatin is not causing sex differences. a**, Adult flies were exposed to control (sucrose) or damage-inducing (DSS) diets. The effect of Y chromosome size on sex differences in intestinal stem cell (ISC) proliferation was quantified by counting the number of mitoses. Representative midguts are shown, DAPI (in blue) and intestinal progenitor marker esg>GFP (in green). *P* values from one-sided Kruskal–Wallis tests are (NS) $P > 0.99$, ***$P < 0.0001$. Scale bars, 60 μm. **b**, Hyperplasic tumour frequency (identified by the accumulation of GFP-positive cells) resulting from adult progenitor-driven *Notch* downregulation and its modulation by the presence and the size of the Y chromosome. Representative confocal images

show intestinal progenitor accumulation, DAPI (in blue) and esg>GFP (in green). *P* values from one-sided Kruskal–Wallis tests are (NS) $P > 0.99$, ***$P < 0.0001$, **$P = 0.0059$. Scale bars, 60 μm. **c**–**e**, Impact of the Y chromosome length on the sexual dimorphisms in pupal volume (**c**), adult weight (**d**) and starvation resistance (**e**). In **c** and **d**, *P* values from one-sided Kruskal–Wallis tests are (NS) $P > 0.99$, ***$P < 0.0001$. In **e**, *P* values from log-rank tests for starvation experiments are (NS) $P > 0.33$. Boxplots display the minimum, the maximum, the sample median and the first and third quartiles. In all panels, *n* = number of flies. Asterisks highlighting comparisons across sexes are displayed in grey boxes; those highlighting comparisons within male datasets are displayed in blue boxes.

the longevity gap which has been recorded in a wide spectrum of species. Epigenetic effects of the Y chromosome have been documented in different conditions[51,52] but, in most cases, Y impacts are revealed only in mutant contexts (as in Fig. 2e) or with transgenic constructs (as in Fig. 2a–d). Although the effects of the Y chromosome on the epigenome were observed in *D. melanogaster* almost 100 years ago[53] and have been studied extensively, the physiological relevance of this

mechanism to sex-specific differences has remained unclear[16], until the work of ref. 12. In this recent work, flies with different sex chromosome karyotypes (XXY females; X0 and XYY males) were generated and it was observed that the number of Y chromosomes correlated with the average lifespan. However, the experimental design could not completely discriminate between genetic background or Y chromosome effects. Our results using isogenic lines demonstrate that the

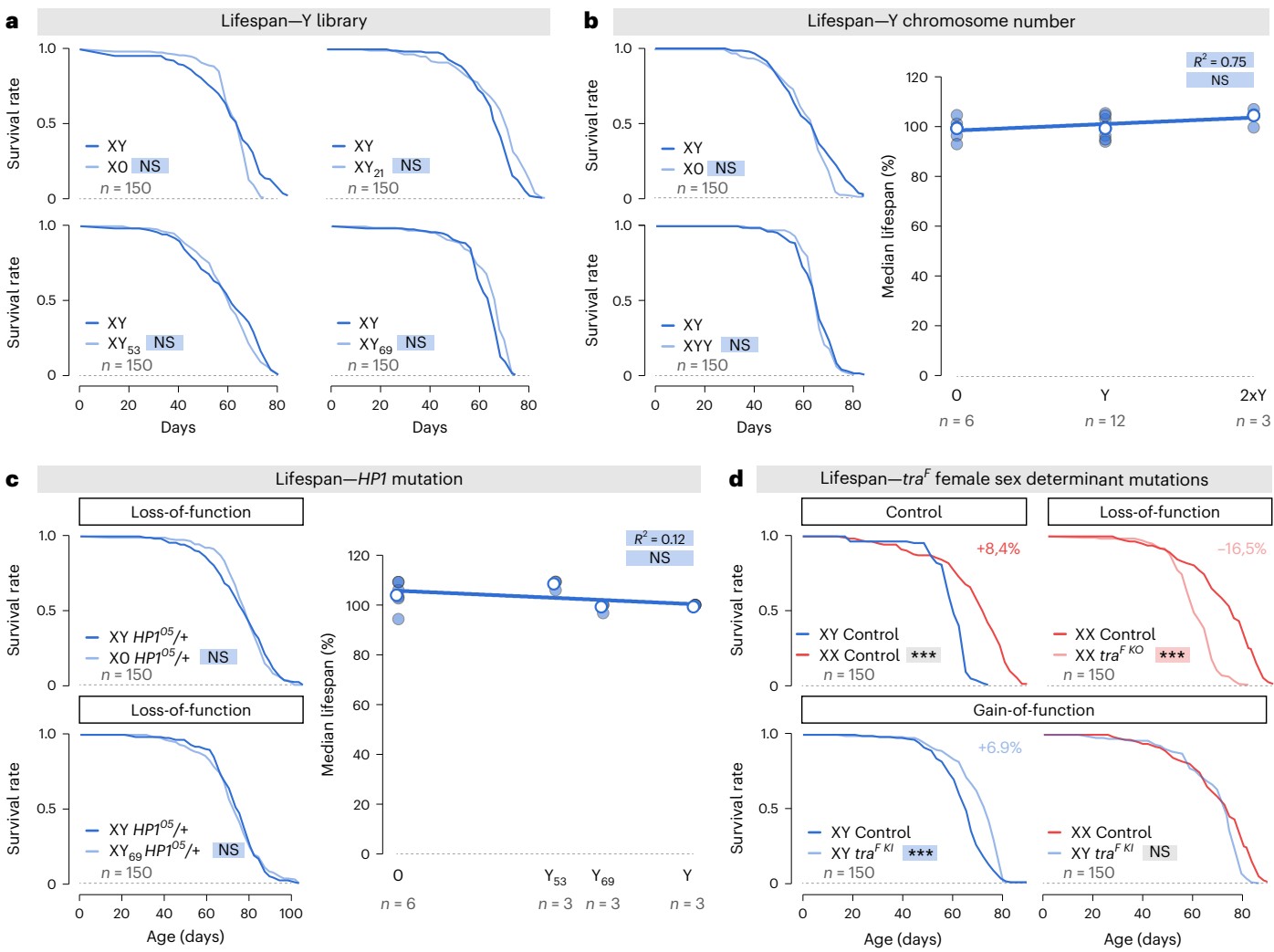

**Fig. 4 | The Y chromosome heterochromatin is not toxic and does not contribute to sex-specific differences in longevity. a**, Lifespan of males carrying Y chromosomes of different sizes, compared to control XY males. $P$ values from log-rank tests for lifespan experiments are (NS) $P > 0.21$. **b**, The lifespan of males with 0, 1 or 2 Y chromosomes. On the left, $P$ values from log-rank tests for lifespan experiments are (NS) $P > 0.44$. On the right, $P$ value and $R^2$ from simple linear regression, (NS) $P = 0.12$. **c**, Effect of Y chromosome size on longevity in males missing one copy of *HP1*. On the left, $P$ values from log-rank tests for lifespan experiments are (NS) $P > 0.55$. On the right, $P$ value and

$R^2$ from simple linear regression, (NS) $P = 0.21$. **d**, Impact of *transformer$^F$* (*tra$^F$*) on the male–female difference in lifespan. The lifespan of females mutated for *tra$^F$* (*tra$^{F KO}$*) and constitutively feminized males carrying a *tra$^F$* knockin allele (*tra$^{F KI}$*). $P$ values from log-rank tests for lifespan experiments are (NS) $P = 0.43$, ***$P < 0.0001$. In all panels, $n$ = number of flies, except for the linear regressions in **b** and **c** where $n$ indicates the number of repeats (each repeat is composed of 150 flies). Asterisks highlighting comparisons across sexes are displayed in grey boxes; those highlighting comparisons within female and male datasets are displayed in red and blue boxes, respectively. See also Extended Data Fig. 3.

reported association was probably not mediated by the quantity of Y chromosome heterochromatin.

Following this observation, all the negative impacts of the Y chromosome on the genome-wide heterochromatin landscape were linked under the toxic Y idea[3,5,11–14]. The toxic Y model postulates that the accumulation of deleterious mutations and repetitive elements on the Y chromosome lowers the survival of the XY heterogametic sex. In this model, the effects depend exclusively on the size of the non-recombining region in the Y chromosomes. Thus, the toxic Y model specifically predicts lower male survival with increasing size of the Y chromosome. In *D. melanogaster*, the Y chromosome represents 13% of the genome (Extended Data Fig. 1g). In studies, where Y chromosome size has been correlated with male survival across hundreds of species of birds, mammals, reptiles and amphibians, all the Y chromosomes considered are smaller than 5% of the genome[4–7]. So, the lack of Y chromosome effect in *D. melanogaster* is a strong argument to discard the potential toxic Y effect in most, if not all, other species where the

Y chromosome is relatively much smaller. For comparison, in XYY male flies, the Y chromosome material represents 26% of the entire genome (Extended Data Fig. 1g). In humans, where the Y chromosome is one of the smallest chromosomes, it represents less than 2% of the genome. It is possible that animals have evolved to express heterochromatin factors at levels that can mitigate the potentially detrimental effects of Y chromosome heterochromatin. Artificially decreasing the level of HP1 by 50% is enough to detect effects at the molecular level (Fig. 2e) but not at any other level: at the cellular, tissue or organismal (Fig. 4c). There is a clear uncoupling between the effects of the Y chromosome on constitutive heterochromatin and its dispensability for biologically relevant sex differences.

Interestingly, two recent studies reported, like us, that the Y chromosome does not affect longevity. Others[54] explored the natural variation of the Y chromosome size in *D. pseudoobscura*. Although males with a larger Y chromosome showed slightly increased levels of transposable element expression in this species, and more so in old

males, these differences were relatively minor and did not result in faster ageing. Investigating sex-specific nutritional requirements for adult lifespan in *D. melanogaster*, a previous study[55] performed longevity experiments on diverse diets. The authors did not observe any differences between XY and X0 males or between XX and XXY females. It is important to emphasize that, unlike ours, both studies could not control for the genetic background of the sex chromosomes. Thus, the lack of effect of the Y chromosome in longevity appears quite robust and is found in the *Dahomey* background[55], the *Canton-S* background (our study, Fig. 4a), the *HP1[05]* mutant conditions (our study, Fig. 4c), in *D. pseudoobscura*[54] and using manipulation of sex chromosome karyotype at the whole-chromosome level (XXY females; X0 and XYY males) (ref. 55 and our study, Fig. 4b) or polymorphic Y of different sizes (ref. 54 and our study, Fig. 4a).

Contrary to the toxic Y hypothesis, we discovered that sex-specific differences in longevity are determined by phenotypic sex (Fig. 4d). Genetic manipulations of a single master-switch gene, *tra[F]*, allowed bidirectional transformation in longevity in both sexes. This effect was independent of the presence of a Y chromosome. Our discovery of this sex-specific contribution to lifespan opens up new lines of research and raises the possibility that other sex determinants play equivalent roles in other species. Remarkably, a recent study reported an effect of *tra[F]* on sex differences in longevity in the context of lifespan extension in response to rapamycin treatment[56]. In the nematode *Caenorhabditis elegans*, the terminal effector of the sex-determination pathway, Transformer protein 1 (Tra-1), has also been linked to the lifespan extension of hermaphrodites compared to males[44].

The cellular and molecular mechanisms underlying positive *tra[F]* effect on longevity deserve further investigation. The effect of *tra[F]* is unlikely to involve sex organs and sex organ-derived hormones as feminized *tra[F]*-expressing males and masculinized *tra[F]* knockout females have atrophic gonads rather than ovaries or testes, respectively[57]. Even so, these flies have a longevity trait that is consistent with their physiological sex as assessed by sex-specific traits. In future, it will be of interest to explore the relative contribution of the intrinsic sexual identity of specific somatic organs. In addition, it will be important to determine whether female-specific lifespan extension is encoded during a critical developmental period and, if not, whether sex-specific longevity can be reprogrammed during adult life.

## Methods

### Fly strains and media

**UAS transgenes.** The following were used in this study: *UAS-Cas9* (BDSC: 67086, in *w\** background), *UAS-sgRNA x14* (this study, in *y¹,w¹¹¹⁸* background), *UAS-sgRNA x27* (this study, in *y¹,w¹¹¹⁸* background), *UAS-sgRNA x66* (this study, in *y¹,w¹¹¹⁸* background), *UAS-sgRNA x223* (this study, in *y¹,w¹¹¹⁸* background), *UAS-Notch[RNAi]* (VDRC GD 27229, in *w¹¹¹⁸* background) and *Su(var)3−9 11 kb* (generated in ref. 36, gift from G. Reuter, in *w¹¹¹⁸* background).

**Mutants.** The following were used in this study: *Su(var)3-9²* (BDSC: 6210, allele ID: FBal0016558, in *In(1)w[m4]* background), *Su(var)205⁵* (BDSC: 6234, allele ID: FBal0016507, in *In(1)w[m4h]* background), *traKO* (BDSC: 67412, allele ID: FBti0186559, in *w¹¹¹⁸* background) and *tra[F]* (generated in ref. 57, in *w¹¹¹⁸* background).

**Gal4 drivers.** The following were used in this study: *nubbin-Gal4* (BDSC: 67086, allele ID: FBti0016825, in *w\** background), *eyeless-Gal4* (gift from P. Meier, generated in ref. 58), *bam-Gal4* (gift from M. Amoyel, generated in ref. 59) and *esg-Gal4[NP7397], UAS-GFP, Tub-Gal80[TS]* chromosome (gift from J. de Navascués, published in ref. 60).

**Reporters.** The following were used in this study: *w[118E-25]* (BDSC: 84091, allele ID: FBti0018330, in *y¹,w[67c23]* background), *w[118E-10]* (BDSC: 84108, allele ID: FBti0016149, in *y¹,w[67c23]* background) and *LacZ[In(3L)BL1]* (BDSC: 57370, allele ID: FBti0015491, in *y¹,w\*; In(3 L)BL1* background).

**Sex chromosomes.** The following were used in this study: *Y[Bar]* (BDSC: 81622, allele ID: FBab0029200, in *w¹¹¹⁸* background), *Y[RFP]* (BDSC: 78567, allele ID: FBti0199495, in *w¹¹¹⁸* background), *Y[Bar+yellow]* (BDSC: 3707, allele ID: FBab0003169, in *y¹,w[a],Ste¹* background), *Y₂₁* (this study), *Y₂₆* (this study), *Y₅₃* (this study), *Y₆₉* (this study), *Y₇₂* (this study), *C(1)DX* (BDSC: 64, allele ID: FBab0000080, in *lz³* background), *C(1)M4* (BDSC: 1999, allele ID: FBab0000083, in *y²* background), *C(1)RM* (BDSC: 4248, allele ID: FBab0000088, in *y¹,pn¹,v¹* background), *C(1,Y)1* (BDSC: 4248, allele ID: FBab0010396, in *y¹,Bar¹* background), *GFP[1A]* (BDSC: 29905, allele ID: FBti0128225, in *w¹¹¹⁸* background), *GFP[11F]* (BDSC: 81134, allele ID: FBti0200716, in *y¹,w\** background) and *GFP[20F]* (BDSC: 24572, allele ID: FBti0078187, in *w¹¹¹⁸* background).

Animals were reared at 25 °C on fly food containing: 10 g of agar, 83 g of corn flour, 60 g of white sugar, 34 g of dry yeast and 3.75 g of Moldex (per litre, diluted in ethanol).

### Plasmids and generation of transgenic lines

**Generation of the UAS-sgRNA transgenes targeting the Y chromosome.** The sgRNAs were designed using the following websites: http://targetfinder.flycrispr.neuro.brown.edu/ and http://www.flyrnai.org/evaluateCrispr/ and using the improved sequence of the *Drosophila* Y chromosome published previously[61]. The following sgRNAs, targeting Y-specific sequences, were cloned: for UAS-sgRNA ×14: AGTCTCCAGCTATACCACCAGGG (cutting 14 times specific sequences in the *Pp1-Y2* locus), UAS-sgRNA ×27: AGCATCCCATCTGTGGCAGGAGG and ATGGTCTCTCTTCTCCCAAGCGG (cutting 27 times specific sequences in the *FDY* locus), UAS-sgRNA ×66: TAAATTCCACACTCGAACCATGG, TGGCTGGGGCTTGCGGGCGCTGG and GCTTTGCAGCAGTCCGGCTAAGG (cutting 66 times specific sequences in the Y centromeric *18HT* repeats) and for UAS-sgRNA ×223: CTCCCCTAACACGTCCTGCCAGG, TGCTTCGAGGACTTGGCCCGCGG, TGGCTGGGGCTTGCGGGCGCTGG and ACTTTGAACCAAGTATTTAGAGG (cutting 223 times specific sequences localized in the *Su(Ste)* repeats). The different sgRNAs, for each UAS transgene, were assembled from overlapping PCR products. PCRs were performed with Q5 high-fidelity polymerase from New England Biolabs (M0491S). The final PCR product was then cloned into Bbs1-digested pCFD6 vector (Addgene: plasmid no. 73915, generated in ref. 62). The constructs were sequence-verified and transgenic lines were established through ΦC-31 integrase-mediated transformation (Bestgene), using the VK05 (BDSC: 9725) attP site line.

### Generation of the library of Y chromosome deletions

Y truncations were kept as supernumerary Y chromosomes, in stocks with males carrying attached sex chromosomes X^Y (*C(1;Y)1*, BL 4248) and females carrying attached X chromosomes, X^X (*C(1)RM*, BL 4248). In these lines, females are X^XY* and males X^YY*, Y* being a specific Y truncation. Both compound chromosomes (X^X and X^Y) were first backcrossed for eight generations into *Canton-S* genetic stock of the laboratory. Y truncations were generated by crossing a single male, expressing a specific *sgRNA* along with the Cas9 protein in the male germline under the *bam-Gal4* driver, with females carrying attached X chromosomes. Single females from the progeny (carrying the attached X and the truncated Y) were backcrossed with males carrying attached X^Y chromosomes. Since Y truncations are kept as additional Y chromosomes, they can accumulate Y-linked mutations and derived for the original stock. We only used truncation lines constructed recently to avoid the accumulation of these Y-linked mutations. All the experiments are based on the same Y truncation lines that have been genotyped before each experiment. The different Y truncations can very easily be generated, at any time, by crossing our different sgRNA lines, targeting specific Y loci, with a *bam-Gal4, UAS-Cas9* line. The Y₂₁ was generated using *sgRNA x66*, Y₂₆ *sgRNA x27*, Y₅₃ *sgRNA x66*, Y₆₉ *sgRNA x223* and Y₇₂ *sgRNA x14*.

## Genotyping the Y chromosome deletions

The genotyping of partially deleted Y chromosomes was performed by PCR from DNA extracted from single adult males using the OneTaq Quick-Load 2× (BioLabs). Primers amplifying the following Y-specific genes were used: *kl-5* (5′-GCGAGAGCTAGAGGGTTGG-3′ and 5′-TGAGGAGCGACTGAGGATTAAAG-3′), *PRY* (5′-GTCAAA GGAACGGCTTCTTAACT-3′ and 5′-GTGAGCGAGCTGGACTGTT-3′), *kl-3* (5′-CGATGTCATAGTTGGGATAACTGATG-3′ and 5′-ATTATTATT GTTTACTTACTATATTTGTTGAGCAGCC-3′), *kl-2* exon 1 (5′-GGCAA CCAGTGAGAACATCG-3′ and 5′-AATTGCGGGGCTTGTTGAAT-3′), *kl-2* exon 12 (5′-CGCCTCCTCCTTCCCTTATT-3′ and 5′-AAACACT CCGCCCTCCAATA-3′), *Pp1-Y1* (5′-CATCGCTGCTTAGCTGGAAG-3′ and 5′-TCG CCC AGC ATC AAA TAA CG-3′), *ARY* (5′-AATACCC ACCATGATCCAAGAGA-3′ and 5′-ATCGCACACGTACTTGTAGCC-3′), *PprY* exon 1 (5′-TCCGAAGTACAGAAGCCCCTT-3′ and 5′-TAAAC GTCTTCCCTCACCACG-3′), *PprY* exon 5 (5′-ATGTGTTGATGACC GTGACG-3′ and 5′-TGTTCGAGTCGCAATTGTGT-3′), *WDY* (5′-CG CAAAGTCATTCAGCGGAG-3′ and 5′-CCGCTCGGGAGTGTTAAGAA-3′), *FDY* (5′-ACGCTAGTCCAGTGAAAGGC-3′ and 5′-GGTCCCCTGTT GTCACGATT-3′), *Mst77Y* (5′-GGTCTGGAAGAGTTGCCCAA-3′ and 5′-GGGCTTAAGACACCTTGGCT-3′), *Pp1-Y2* (5′-CGCCGT CTCAAGCAGCTTAAT-3′ and 5′-GTTGCGACATAAAAACTTCCCG-3′), *ORY* (5′-GGTTAGCGGGAGAAGTTGTGG-3′ and 5′-GAAGCCATT TTGCTCATAGCATC-3′) and *CCY* (5′-CTGCATATTCGCCTGAAATGGG-3′ and 5′-TCGGATTGTTTGCATAGCTCAT-3′).

## Flow cytometry

Our experimental procedure was adapted from refs. 63,64. In brief, 50 fly heads were collected into a Dounce tissue grinder on ice with 500 µl of ice-cold Galbraith buffer ($4.26 \text{ g l}^{-1}$ of $MgCl_2$, $8.84 \text{ g l}^{-1}$ of sodium citrate, $4.2 \text{ g l}^{-1}$ of MOPS, 0.1% Triton X-100, $20 \text{ µg ml}^{-1}$ of RNAse A, pH 7.2). Heads were crushed 20 times with a pestle and transferred into 1.5 ml tubes on ice. Between samples, the Dounce was washed with deionized water. Samples were filtered through a 70 µm nylon mesh to remove bigger debris and the cell suspension was collected into a 5 ml round-bottomed tube on ice. DNA was marked with $40 \text{ µg l}^{-1}$ of propidium iodide mixed and incubated at 4 °C for 2 h in the dark. Cells were analysed on the BD LSRFortessa Cell Analyzer.

## Karyotypes

Brains from third-instar larvae were dissected in 0.7% NaCl, treated with 0.5% Na citrate and fixed for 10–20 s in acetic acid/methanol/ water (11:11:1) at room temperature. Four brains were squashed in 45% acetic acid under a siliconized coverslip and frozen in liquid nitrogen. The coverslip was then removed using razor blade. Samples were air-dried, stained with Orcein staining solution (Jeulin) for 15 min, washed with 45% acetic acid and placed under a coverslip with a drop of 45% acetic acid.

## Immunofluorescence on larval tissues

Larval tissues were dissected in PBS, fixed in 4% formaldehyde (Polyscience) in PBS for 30 min at room temperature and then washed several times in PBS containing 0,3% Triton X-100 (PBT). They were then blocked into PBT + 10% fetal bovine serum. Primary antibodies were incubated overnight at 4 °C. After several washes, secondary antibodies were incubated for 2 h at room temperature. After DAPI staining, dissected tissues were mounted into Vectashield (Vector). Fluorescence images were acquired using a Leica SP5 DS confocal microscope.

The following antibodies were used: mouse anti-Phospho-gammaHis2Av (1/250) (DSHB), rabbit anti-Phospho-histone H3 (Ser10) (1/500) (9701 Cell Signaling), chicken anti-GFP (1/10,000) (ab13970 Abcam), mouse anti-Histone H3 (di-methyl K9) (1/500) (ab1220 Abcam), rabbit anti-Histone H3 (tri-methyl K9) (1/1,000) (ab8898 Abcam), rabbit anti-Stellate (1/1,000) (gift from W. E. Theurkauf[26]).

## Fluorescent in situ hybridization

FISH was performed on partially dissected tissues (brain or imaginal discs) in PBS 1× on ice, fixed in 4% formaldehyde in PBS + 0.1% Tween 20 (PBTw) for 20 min at room temperature on a rotating wheel and washed three times in PBTw. Tissues were incubated with RNAse A at 200 µg in PBTw for 2 h and then incubated for 1 h in PBT at room temperature on a rotating wheel. Tissues were transferred in series of solutions with 80% PBT + 20% prehybridization buffer (PHB) (50% formamide, 2× SSC, 0.1% Tween 20), 50% PBT + 50% PHB, 20% PBT + 80% PHB and finally 100% PHB for 20 min each. Larval DNA was denatured by incubating the tissue in PHB for 15 min at 80 °C. Fluorescently labelled probes (1 µM) were denatured in PHB for 10 min at 80 °C and added to the tissues without cooling for overnight incubation at 37 °C under gentle agitation. Posthybridization washes were made by passing under strong agitation through series of solutions (20 min each) of 50% formamide + 2× SSC twice at 37 °C, then 40% formamide + 2× SSC at 37 °C, 30% formamide + 70% PBTw at 37 °C, 20% formamide + 80% PBTw at 37 °C, 10% formamide + 90% PBTw at RT, 100% PBTw at room temperature and finally 100% PBT at RT. After DAPI staining and finer dissection, tissues were mounted into Vectashield (Vector). Fluorescence images were acquired using a Leica SP5 DS confocal microscope. The following probes were used YS22 AlexaFluor488-(AATAAAC)×6 to label the Y chromosome short arm and YL10 Cy3-(AATAGAC)×6 for the Y chromosome long arm[24].

## Fertility tests

For fertility experiments, male flies were collected and aged for 3 days. They were then mated overnight to Canton-S female flies (one male with four females per vial). Male flies were then removed and single female flies were transferred into individual vials for a 3 day period and progeny was counted.

## Quantification of the rDNA locus presence

Presence of the rDNA locus on modified Y chromosomes was evaluated using the *C(1)DX* compound chromosome. *C(1)DX* are attached X chromosomes lacking the rDNA locus. To be viable, *C(1)DX* females must carry a Y chromosome with an intact rDNA locus. The Y chromosome deletions of the collection were tested by crossing *C(1)DX* females with X^YY* males (Y* being a specific Y chromosome truncation) and female survival was determined by measuring the number of females in the progeny/number of males.

## Eye pigmentation

We performed eye pigmentation quantification as previously described[65]. In brief, five groups of five males were homogenized using a TissueLyser (Qiagen) in 500 µl of 0.01 M HCl in ethanol. Homogenates were incubated overnight at 4 °C, warmed at 50 °C for 5 min and clarified by centrifugation. Optical density of the supernatant was measured at 480 nm.

## X-Gal staining

After heat shock (37 °C for 15 min) and recovery for 1 h at room temperature, Malpighian tubules were stained for β-galactosidase activity using β-Gal Staining Kit (ThermoFisher K146501). In brief, tissues were dissected in PBS, fixed in 2% formaldehyde and 0.2% glutaraldehyde for 30 min at room temperature. After two washes in PBS, tissues were incubated with X-Gal overnight at 37 °C. They were rinsed three times in PBS and stained for DAPI before mounting into Vectashield (Vector). Images were obtained with the digital camera DFC7000T (Leica).

## DSS-induced regeneration

For damage-induced regeneration assays, virgin flies were collected over 72 h at 18 °C and were then shifted to 29 °C for 7 days on standard media. Flies were then transferred into an empty vial containing a piece of 3.75 × 2.5 cm² paper. A total of 500 µl of 5% sucrose solution (control) or 5% sucrose + 3% DSS solution were used to wet the paper,

used as feeding substrate. Flies were transferred into a new vial with fresh feeding paper every day for 3 days before dissection.

### Genetically induced tumours

The *esg-Gal4^NP7397^, UAS-GFP, Tub-Gal80^TS^* and *UAS-Notch^RNAi^* (VDRC GD 27229) strain was used to generate midgut tumours[40]. Adult flies were raised at 18 °C before transfer to 29 °C for 15 days. Tumour incidence was determined by counting the presence of GFP expansion and PH3 staining.

### Pupal volume and adult weight

Larvae were synchronized at 24 h after egg deposition and reared under controlled conditions (30 larvae per vial). Pupal volume was measured using ImageJ and calculated by using the formula $4/3 \times \pi \times (\text{length}/2) \times (\text{width}/2)^2$. Groups of five adult flies were weighed on an XPR analytical balance (Mettler Toledo). For a given experiment, all values were normalized to one control condition.

### Starvation

For starvation assays, flies were maintained exactly as in lifespan studies until 14 days of age. Flies were then transferred into vials containing only 1.5% agar with subsequent transfer into fresh vials three times per week. Deaths were scored four times per day. Accidental deaths and escapees were censored and did not exceed 5%.

### Lifespan

For lifespan experiments, all fly stocks (including the Y chromosome truncation lines) used in this study were backcrossed for six or more generations into *Canton-S* genetic stock from the laboratory, except for the *C(1,Y)1* (BL 4248) chromosome that cannot be backcrossed. Lifespan measurements were performed as previously described[66]. In summary, after 4 h of egg-laying, eggs were collected and dispensed into bottles with fly food for development at standard density. After 10 days, newly emerged flies were transferred into new bottles during 2 days for mating. Flies were then anaesthetized with $CO_2$ to select desired genotypes. Flies were transferred into ten tubes as groups of 15 flies per vial (150 flies per condition). Vials were kept horizontally during lifespan experiments and flies were transferred into fresh food every 2 days. For each transfer, deaths, accidental deaths or escapees were scored. A Microsoft Excel sheet, generated by the laboratory of M. Piper (template available at http://piperlab.org/resources/), was used to record, calculate and plot the data.

### Reverse transcription and quantitative PCR

RNA was extracted from five whole flies using TRIzol (Invitrogen). Complementary DNA was synthesized using the iScript cDNA synthesis kit (Bio-Rad) from 500 ng of total RNA. Quantitative PCR was performed by mixing cDNA samples (5 ng) with iTaq Universal SYBR Green Supermix (Bio-Rad, 172-5124) and the relevant primers in 384-well plates. Expression abundance was calculated using a standard curve for each gene and normalized to the expression of the *rp49* control gene. For data display purposes, the median of the expression abundance was arbitrarily set at 100% for young XY males and percentage of that expression is displayed for all sexes and genotypes. The following quantitative PCR primer pairs were used: *copia* (5′-GCATGAGAGGTTTGGCCATATA AGC-3′, 5′-GGCCCACAGACATCTGAGTGTACTACA-3′), *rp49* (5′-CTT CATCCGCCACCAGTC-3′, 5′-CGACGCACTCTGTTGTCG-3′).

### Quantification of the H3K9me2 and H3K9me3 signals

H3K9me3/2 signals were quantified by imaging posterior midguts and tubules at 20× magnification. H3K9me3/2 area was quantified using ImageJ. Threshold was adjusted for the H3K9me3/2 channel (ImageJ function: Image > Adjust > Threshold) to subtract background, then the size of the area above the threshold was considered (ImageJ function: analyse particles). The same analysis was done for the DAPI signal and for every single nucleus, the area of the H3K9me3/2 channel

signal was divided by the total area of the nucleus (DAPI channel). H3K9me3/2 signal intensity was quantified using the same method. Threshold was adjusted for the H3K9me3/2 channel (ImageJ function: Image > Adjust > Threshold) to subtract background, then the average intensity of the signal was measured in every nucleus (ImageJ function: analyse particles). Data were collected from at least 50 nuclei per genotype and are displayed as boxplots showing all data points.

### Statistics and data presentation

All statistical analyses were carried out in GraphPad Prism v.7.04. Comparisons between two genotypes and/or conditions were analysed with the Mann–Whitney–Wilcoxon rank sum test. Multiple comparisons between a single control condition and different genotypes were analysed using one-way non-parametric analysis of variance (ANOVA). These two non-parametric tests do not require the assumption of normal distributions, so no methods were used to determine whether the data met such assumptions. For lifespan experiments, log-rank tests were used. All graphs were generated using GraphPad Prism v.7.04. All confocal and bright field images belonging to the same experiment and displayed together in our figures were acquired using the exact same settings. For visualization purposes, level and channel adjustments were applied using ImageJ to the confocal images shown in the figure panels (the same correction was applied to all images belonging to the same experiment) but all quantitative analyses were carried out on unadjusted raw images or maximum projections.

### Reporting summary

Further information on research design is available in the Nature Portfolio Reporting Summary linked to this article.

## Data availability

All data are available in the main text or Supplementary Information. Materials generated for the study are available from the corresponding authors on request.

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

## Acknowledgements

We thank W. Theurkauf for the anti-Stellate antibody; G. Reuter, T. Jenuwein, P. Meier and M. Amoyel for fly lines; the Bloomington Drosophila Stock Center for providing *Drosophila* fly lines; the iBV platforms—A. Loubat from the flow cytometry platform, B. Monterroso and S. B. Aicha from the imaging facility and A. Mencser for fly food; F. Besse and S. Merabet for comments; and all the members of the B.H. laboratory for discussions and comments. This work was supported by the Université Côte d'Azur, CNRS (ATIP-Avenir programme), INSERM, European Research Council (ERC starting grant CellSex, grant no. ERC-2019-STG 850934) and the French Government (National Research Agency, ANR) through the 'Investments for the Future' programmes LABEX SIGNALIFE ANR-11-LABX-0028-01 and IDEX UCAJedi ANR-15-IDEX-01.

## Author contributions

B.H., R.D. and C.C. conceived of the work. B.H., R.D. and C.C. were responsible for methodology. R.D., C.C., C.L. and T.P. undertook investigations. R.D., C.C. and C.L. carried out visualization. B.H. acquired funding and undertook project administration. B.H. and R.D. provided supervision. The original draft was written by B.H.

## Competing interests

The authors declare no competing interests.

## Additional information

**Extended data** is available for this paper at https://doi.org/10.1038/s41559-023-02089-7.

**Correspondence and requests for materials** should be addressed to Rénald Delanoue or Bruno Hudry.

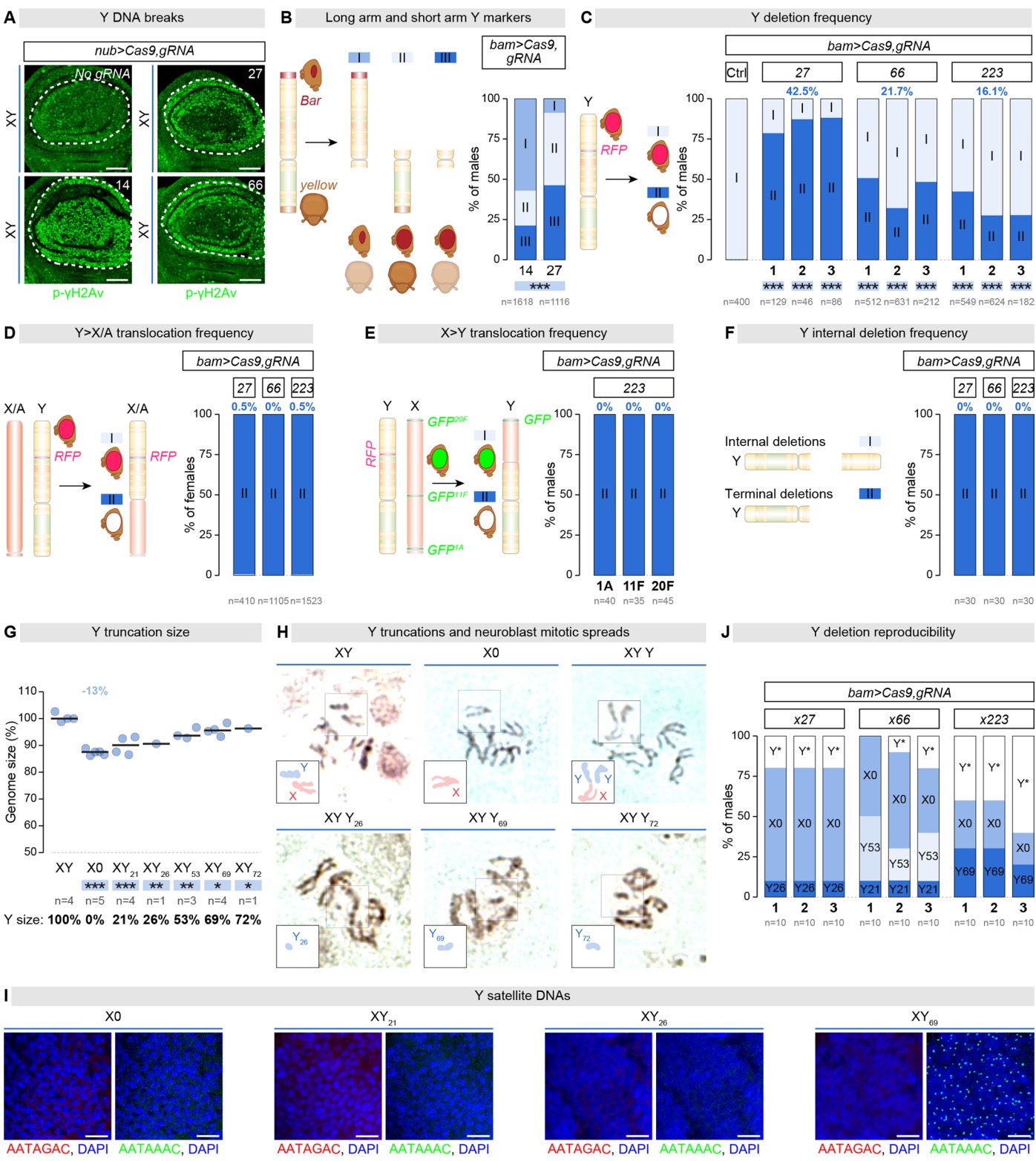

**Extended Data Fig. 1 | See next page for caption.**

**Extended Data Fig. 1 | Production of polymorphic Y chromosomes of different sizes using CRISPR. (a)** Wing discs of third-instar expressing Cas9 and the indicated *sgRNAs* under the control of the *nub-Gal4* driver (expressed in the wing pouch, dashed lines) and stained for the DNA double-strand breaks marker p-γH2Av (green). Scale bars, 50 μm. Immunohistochemical analyses were repeated three independent times. **(b)** Percentage of males expressing the Y-linked markers: *Bar* and *yellow* in the progeny of males expressing Cas9 and a *sgRNA* under the control of the *bam-Gal4* driver. P-value from one-sided t test is ***p < 0.0001. **(c)** Percentage of males expressing the Y-linked RFP (I) or not (II) in the progeny of males expressing Cas9 and the indicated *sgRNAs* under the control of the *bam-Gal4* driver. P-values from one-sided ANOVA are ***p < 0.0001. **(d)** Percentage of females expressing the RFP translocated from the Y chromosome (I) or not (II) in the progeny of males expressing Cas9 and the indicated *sgRNAs* under the control of the *bam-Gal4* driver. **(e)** Percentage of males, who have lost the Y-linked RFP marker, expressing GFP transgenes translocated from the X chromosome (I) or not (II) in the progeny of males expressing Cas9 and the indicated *sgRNAs* under the control of the *bam-Gal4* driver. **(f)** Percentage of males carrying internal (I) versus terminal (II) deletions, determined by PCR genotyping analyses of the Y-linked coding genes of males,

who have lost the Y-linked RFP marker, in the progeny of males expressing Cas9 and the indicated *sgRNAs* under the control of the *bam-Gal4* driver. **(g)** Diploid genome size estimations of the different Y truncations obtained from flow cytometry, compared to wild-type XY and X0 males. The percentage of the Y chromosome remaining for each deletion is displayed at the bottom. P-values from one-sided ANOVA are *p < 0.03, **p < 0.0098, ***p < 0.0001. **(h)** Karyotypes from larval neuroblasts stained with orcein for the indicated lines. Y chromosomes are indicated in blue, and X chromosomes in red. **(i)** DNA-FISH analysis of larval brains of males carrying the indicated Y chromosome. Probe sequences of the Y satellite DNA used are specified by the coloured text, DAPI (in blue). Scale bars, 10 μm. FISH analyses were repeated three independent times. **(j)** Percentage of males carrying a specific deletion from our Y library, unanticipated deletions (Y*), or a complete Y deletion (X0), determined by PCR genotyping analyses of the 13 Y-linked coding genes of males, who have lost the Y-linked RFP marker, in the progeny of males expressing Cas9 and the indicated *sgRNA* under the control of the *bam-Gal4* driver. In all panels, n = number of flies, except in panel (g) where n indicates the number of repeats (each repeats composed of 50 fly heads). Asterisks highlighting significant comparisons within male datasets are displayed in blue boxes.

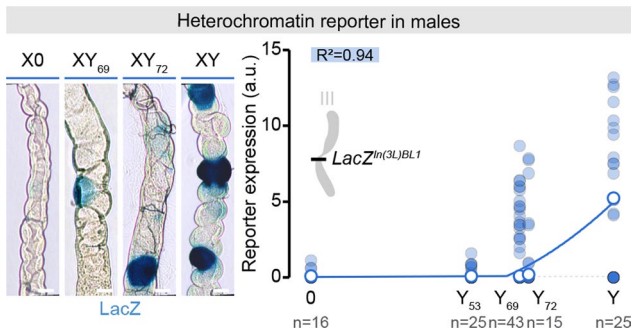

**Extended Data Fig. 2 | The Y chromosome has size-dependent effects on heterochromatin in *trans*.** Effect of Y chromosome length on the expression of the LacZ-based heterochromatin reporter *In(3 L)BL1* in male Malpighian tubules. Representative pictures show LacZ's expression in blue. Scale bars, 10 μm. n = number of tubules (at least 150 cells were counted by tubule). $R^2$ from nonlinear regression, using a polynomial equation.

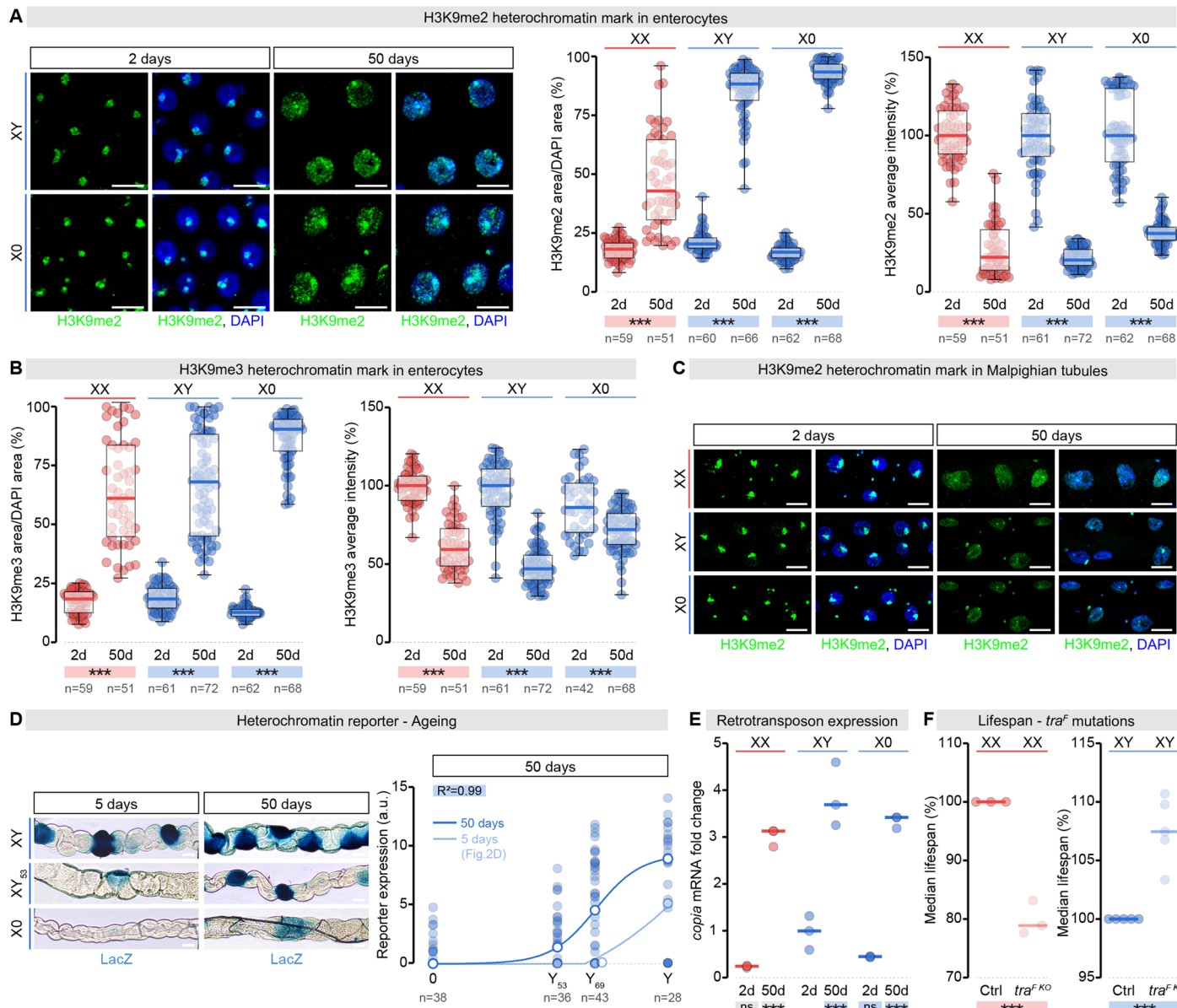

**Extended Data Fig. 3 | The Y chromosome heterochromatin is not toxic and does not contribute to the sex gap in longevity. (a)** Impact of the Y chromosome on the age-related loss and dispersion of the heterochromatin marker H3K9me2 in adult intestinal enterocytes was quantified by measuring the average intensity of the H3K9me2 signal by nucleus, as well as the area of the signal compared to the area of the nucleus. Representative pictures from young (2 days) and old (50 days) individuals showing H3K9me2 signal in green and DAPI in blue. Scale bars, 10 μm. n = number of cells. P-values from one-sided Mann–Whitney-Wilcoxon tests are ***p < 0.0001. **(b)** The impact of the Y chromosome on the age-related loss and dispersion of the heterochromatin marker H3K9me3 in adult intestinal enterocytes was quantified by measuring the average intensity of the H3K9me3 signal by nucleus, as well as the area of the signal compared to the area of the nucleus. n = number of cells. P-values from one-sided Mann–Whitney-Wilcoxon tests are ***p < 0.0001. **(c)** Impact of the Y chromosome on the age-related loss and dispersion of the heterochromatin marker H3K9me2 in adult Malpighian tubule cells. Representative pictures from young (2 days) and old (50 days) individuals showing H3K9me2 signal in green and DAPI in

blue. Scale bars, 10 μm. **(d)** Effect of Y chromosome length on expression of the LacZ-based heterochromatin reporter *In(3 L)BL1* in male Malpighian tubules during ageing. Representative pictures from young (5 days) and old (50 days) males showing LacZ expression in blue. Scale bars, 10 μm. n = number of tubules tested (at least 150 cells where counted by tubule). R² from nonlinear regressions, using sigmoidal dose–response equations. **(e)** RT-qPCR analyses of the *copia* retrotransposon expression between young (2 days) and old (50 days) females, males, and X0 males. n = number of repeats (each repeat is composed of 5 flies). P-values from one-sided t tests are (ns) p > 0.07, ***p < 0.0001. **(f)** Impact of *transformer^F* (*tra^F*) on lifespan. The lifespan of females mutated for *tra^F* (*tra^FKO*) and constitutively feminized males carrying a *tra^F* knock-in allele (*tra^FKI*). n = number of repeats (each repeat is composed of 150 flies). P-values from one-sided t tests are ***p < 0.0001. In all panels, boxplots display the minimum, the maximum, the sample median, and the first and third quartiles. Asterisks highlighting comparisons across sexes are displayed in grey boxes; those highlighting comparisons within female and male datasets are displayed in red and blue boxes respectively.

**Extended Data Table 1 | Production of polymorphic Y chromosomes of different sizes using CRISPR**

| Y Genes | 1 | 2 | 3 | 4 | 5 | 6 | 7 | 8 | 9 | 10 | 11 | 12 | 13 | 14 | 15 | 16 | 17 | 18 | 19 | 20 | 21 | 22 | 23 | 24 | 25 | 26 | 27 | 28 | 29 | 30 |
|---|---|---|---|---|---|---|---|---|---|---|---|---|---|---|---|---|---|---|---|---|---|---|---|---|---|---|---|---|---|---|
| *kl-5* | - | - | - | - | - | - | - | - | - | - | - | - | - | - | - | - | - | - | - | - | - | - | - | - | - | - | - | - | - | - |
| *PRY* | - | - | - | - | - | - | - | - | - | - | - | - | - | - | - | - | - | - | - | - | - | - | - | - | - | - | - | - | - | - |
| *kl-3* | - | - | - | - | - | - | - | - | - | - | - | - | - | - | - | - | - | - | - | - | - | - | - | - | - | - | - | - | - | - |
| *kl-2* | - | - | - | - | - | - | - | - | - | - | - | - | - | - | - | - | - | - | - | - | - | - | - | - | - | - | - | - | - | - |
| *ARY* | - | - | - | - | - | - | - | - | - | - | - | - | - | - | - | - | - | - | - | - | - | - | - | - | - | - | - | - | - | - |
| *PprY* | - | - | - | - | - | - | - | - | - | - | - | - | - | - | - | - | - | - | - | - | - | - | - | - | - | - | - | - | - | - |
| *WDY* | - | - | - | - | - | - | - | - | - | - | - | - | - | - | - | - | - | - | - | - | - | - | - | - | - | - | - | - | - | - |
| *Mst77Y* | + | + | + | + | + | - | - | - | - | - | - | + | + | + | - | - | - | - | - | - | + | + | + | + | + | - | - | - | - | - |
| *Pp1-Y2* | - | + | + | + | + | - | - | - | - | - | - | + | + | - | - | - | - | - | - | - | - | + | + | + | + | + | - | - | - | - |
| *ORY* | - | + | + | + | + | - | - | - | - | - | - | + | + | + | - | - | - | - | - | - | - | + | + | + | - | - | - | - | - | - |
| *CCY* | - | + | + | + | + | - | - | - | - | - | - | + | + | + | - | - | - | - | - | - | - | + | + | + | - | - | - | - | - | - |

| Y Genes | 1 | 2 | 3 | 4 | 5 | 6 | 7 | 8 | 9 | 10 | 11 | 12 | 13 | 14 | 15 | 16 | 17 | 18 | 19 | 20 | 21 | 22 | 23 | 24 | 25 | 26 | 27 | 28 | 29 | 30 |
|---|---|---|---|---|---|---|---|---|---|---|---|---|---|---|---|---|---|---|---|---|---|---|---|---|---|---|---|---|---|---|
| *kl-5* | - | - | - | - | - | - | - | - | - | - | - | - | - | - | - | - | - | - | - | - | - | - | - | - | - | - | - | - | - | - |
| *PRY* | - | - | - | - | - | - | - | - | - | - | - | - | - | - | - | - | - | - | - | - | - | - | - | - | - | - | - | - | - | - |
| *kl-3* | - | - | - | - | - | - | - | - | - | - | - | - | - | - | - | - | - | - | - | - | - | - | - | - | - | - | - | - | - | - |
| *kl-2* | - | - | - | - | - | - | - | - | - | - | - | - | - | - | - | - | - | - | - | - | - | - | - | - | - | - | - | - | - | - |
| *ARY* | + | + | + | + | + | - | - | - | - | - | + | + | + | + | + | + | + | - | - | - | + | + | + | + | + | + | - | - | - | - |
| *PprY* | + | + | + | + | + | + | + | - | - | - | + | + | + | + | + | + | + | - | - | - | + | + | + | + | + | + | + | + | - | - |
| *WDY* | + | + | + | + | + | + | + | - | - | - | + | + | + | + | + | + | + | - | - | - | + | + | + | + | + | + | + | + | - | - |
| *Mst77Y* | + | + | + | + | + | + | + | - | - | - | + | + | + | + | + | + | + | - | - | - | + | + | + | + | + | + | + | + | - | - |
| *Pp1-Y2* | + | + | + | - | - | - | - | - | - | - | + | + | + | + | + | - | - | - | - | - | + | + | - | - | - | - | - | - | - | - |
| *ORY* | + | + | + | - | - | - | - | - | - | - | + | + | + | - | - | - | - | - | - | - | + | + | - | - | - | - | - | - | - | - |
| *CCY* | + | + | + | - | - | - | - | - | - | - | + | + | + | - | - | - | - | - | - | - | + | + | - | - | - | - | - | - | - | - |

| Y Genes | 1 | 2 | 3 | 4 | 5 | 6 | 7 | 8 | 9 | 10 | 11 | 12 | 13 | 14 | 15 | 16 | 17 | 18 | 19 | 20 | 21 | 22 | 23 | 24 | 25 | 26 | 27 | 28 | 29 | 30 |
|---|---|---|---|---|---|---|---|---|---|---|---|---|---|---|---|---|---|---|---|---|---|---|---|---|---|---|---|---|---|---|
| *kl-5* | - | - | - | - | - | - | - | - | - | - | - | - | - | - | - | - | - | - | - | - | - | - | - | - | - | - | - | - | - | - |
| *PRY* | - | - | - | - | - | - | - | - | - | - | - | - | - | - | - | - | - | - | - | - | - | - | - | - | - | - | - | - | - | - |
| *kl-3* | - | - | - | - | - | - | - | - | - | - | - | - | - | - | - | - | - | - | - | - | - | - | - | - | - | - | - | - | - | - |
| *kl-2* | - | - | - | - | - | - | - | - | - | - | - | - | - | - | - | - | - | - | - | - | - | - | - | - | - | - | - | - | - | - |
| *ARY* | - | + | + | - | - | - | - | - | - | - | - | - | + | - | - | - | - | - | - | - | - | - | + | - | - | - | - | - | - | - |
| *PprY* | - | + | + | - | - | - | - | - | - | - | - | - | + | - | - | - | - | - | - | - | - | - | + | - | - | - | - | - | - | - |
| *WDY* | - | + | + | - | - | - | - | - | - | - | - | - | + | - | - | - | - | - | - | - | - | - | + | - | - | - | - | - | - | - |
| *Mst77Y* | + | + | + | - | - | - | - | - | - | - | + | + | + | - | - | - | - | - | - | - | + | + | + | - | - | - | - | - | - | - |
| *Pp1-Y2* | + | + | + | - | - | - | - | - | - | - | + | + | + | - | - | - | - | - | - | - | + | + | + | - | - | - | - | - | - | - |
| *ORY* | - | + | + | - | - | - | - | - | - | - | - | - | + | - | - | - | - | - | - | - | - | + | + | - | - | - | - | - | - | - |
| *CCY* | - | + | + | - | - | - | - | - | - | - | - | - | + | - | - | - | - | - | - | - | - | + | + | - | - | - | - | - | - | - |

PCR genotyping analyses of the Y-linked coding genes on 90 Y deletions presented in Extended Data Fig. 1f. Present and deleted genes are symbolized by green and red boxes, respectively.

Hudry bruno

# Reporting Summary

## Statistics

For all statistical analyses, confirm that the following items are present in the figure legend, table legend, main text, or Methods section.

| n/a | Confirmed | |
|---|---|---|
| ☐ | ☒ | The exact sample size (*n*) for each experimental group/condition, given as a discrete number and unit of measurement |
| ☐ | ☒ | A statement on whether measurements were taken from distinct samples or whether the same sample was measured repeatedly |
| ☐ | ☒ | The statistical test(s) used AND whether they are one- or two-sided *Only common tests should be described solely by name; describe more complex techniques in the Methods section.* |
| ☒ | ☐ | A description of all covariates tested |
| ☐ | ☒ | A description of any assumptions or corrections, such as tests of normality and adjustment for multiple comparisons |
| ☒ | ☐ | A full description of the statistical parameters including central tendency (e.g. means) or other basic estimates (e.g. regression coefficient) AND variation (e.g. standard deviation) or associated estimates of uncertainty (e.g. confidence intervals) |
| ☐ | ☒ | For null hypothesis testing, the test statistic (e.g. *F*, *t*, *r*) with confidence intervals, effect sizes, degrees of freedom and *P* value noted *Give P values as exact values whenever suitable.* |
| ☒ | ☐ | For Bayesian analysis, information on the choice of priors and Markov chain Monte Carlo settings |
| ☒ | ☐ | For hierarchical and complex designs, identification of the appropriate level for tests and full reporting of outcomes |
| ☒ | ☐ | Estimates of effect sizes (e.g. Cohen's *d*, Pearson's *r*), indicating how they were calculated |

*Our web collection on statistics for biologists contains articles on many of the points above.*

## Software and code

Policy information about availability of computer code

| Data collection | *Provide a description of all commercial, open source and custom code used to collect the data in this study, specifying the version used OR state that no software was used.* |
|---|---|
| Data analysis | All statistical analyses were carried out using ImageJ (2015), Microsoft Excel (version 16.16.27), and GraphPad Prism 9.2.0 (283). |

For manuscripts utilizing custom algorithms or software that are central to the research but not yet described in published literature, software must be made available to editors and reviewers. We strongly encourage code deposition in a community repository (e.g. GitHub). See the Nature Portfolio guidelines for submitting code & software for further information.

## Data

Policy information about availability of data

All manuscripts must include a data availability statement. This statement should provide the following information, where applicable:

- Accession codes, unique identifiers, or web links for publicly available datasets
- A description of any restrictions on data availability
- For clinical datasets or third party data, please ensure that the statement adheres to our policy

All data is available in the main text or the supplementary data. Materials generated for the study are available from the corresponding authors on request.

## Human research participants

Policy information about studies involving human research participants and Sex and Gender in Research.

| | |
|---|---|
| Reporting on sex and gender | n/a |
| Population characteristics | *Describe the covariate-relevant population characteristics of the human research participants (e.g. age, genotypic information, past and current diagnosis and treatment categories). If you filled out the behavioural & social sciences study design questions and have nothing to add here, write "See above."* |
| Recruitment | *Describe how participants were recruited. Outline any potential self-selection bias or other biases that may be present and how these are likely to impact results.* |
| Ethics oversight | *Identify the organization(s) that approved the study protocol.* |

Note that full information on the approval of the study protocol must also be provided in the manuscript.

# Field-specific reporting

Please select the one below that is the best fit for your research. If you are not sure, read the appropriate sections before making your selection.

☒ Life sciences  ☐ Behavioural & social sciences  ☐ Ecological, evolutionary & environmental sciences

For a reference copy of the document with all sections, see nature.com/documents/nr-reporting-summary-flat.pdf

# Life sciences study design

All studies must disclose on these points even when the disclosure is negative.

| | |
|---|---|
| Sample size | No a priori sample-size calculation was performed, sample size was set according to the reproducibility of each experiment. ARRIVE guidelines had been followed for the study and the maximum number of replicates were used for each experiment above which additional replicates did not alter the statistical significance. |
| Data exclusions | No data were excluded from the analyses. |
| Replication | Each unique experiments were repeated at least three independent times (n). "n" refers to the number of biological replicates for each experimental groups. The number of technical replicates, the experimental units, and number of experimental units allocated to each group are indicated for all experiments in the Figures and/or in the Figure legends. All attempts at replication were successful. |
| Randomization | Samples were randomly selected for analysis and were randomly allocated into experimental groups. |
| Blinding | All experiments were conducted single-blind. Each experimental group were given numbers prior to dissection and analysis. Only after data were recorded experimental numbers were brought with the genotypes/treatment groups. |

# Reporting for specific materials, systems and methods

We require information from authors about some types of materials, experimental systems and methods used in many studies. Here, indicate whether each material, system or method listed is relevant to your study. If you are not sure if a list item applies to your research, read the appropriate section before selecting a response.

### Materials & experimental systems

| n/a | Involved in the study |
|---|---|
| ☐ ☒ | Antibodies |
| ☒ ☐ | Eukaryotic cell lines |
| ☒ ☐ | Palaeontology and archaeology |
| ☐ ☒ | Animals and other organisms |
| ☒ ☐ | Clinical data |
| ☒ ☐ | Dual use research of concern |

### Methods

| n/a | Involved in the study |
|---|---|
| ☒ ☐ | ChIP-seq |
| ☒ ☐ | Flow cytometry |
| ☒ ☐ | MRI-based neuroimaging |

# Antibodies

| | |
|---|---|
| Antibodies used | The following antibodies were used: mouse anti-Phospho-gammaHis2Av (1/250) (DSHB), rabbit anti-Phospho-histone H3 (Ser10) (1/500) (9701 Cell Signaling), chicken anti-GFP (1/10000) (ab13970 Abcam), mouse anti-Histone H3 (di-methyl K9) (1/500) (ab1220 Abcam), rabbit anti-Histone H3 (tri-methyl K9) (1/1000) (ab8898 Abcam), rabbit anti-Stellate (1/1000) (gift from W.E. Theurkauf26). |
| Validation | mouse anti-Phospho-gammaHis2Av (1/250) (DSHB): validation in The development of a monoclonal antibody recognizing the Drosophila melanogaster phosphorylated histone H2A variant (γ-H2AV). Hawley RS. G3 (Bethesda, Md.) 3.9 (2013 Sep 4): 1539-43. Rabbit anti-phospho-histone H3 Ser10 (9701L, Cell Signalling Technology, validation: Cell Signalling Technology for W, IHC-P, IF-IC, Reacts with-D. melanogaster) <br> chicken anti-GFP (1/10000) (ab13970 Abcam, validation: Abcam suitable for ELISA, IHC-Fr, ICC, IHC-P, IP, WB, IHC-FoFr, IHC-FrFl, Electron Microscopy, Reacts with: Species independent) <br> mouse anti-Histone H3 (di-methyl K9) (1/500) (ab1220 Abcam, validation: Abcam suitable for: ICC/IF, WB, ELISA, IHC-P, ChIP, reacts with: Cow, Human, Arabidopsis thaliana, Drosophila melanogaster, Rice, Recombinant fragment) <br> rabbit anti-Histone H3 (tri-methyl K9) (1/1000) (ab8898 Abcam, validation: Abcam suitable for: WB, IHC-P, ICC, ChIP, reacts with: Mouse, Cow, Human) <br> rabbit anti-Stellate (1/1000) (gift from W.E. Theurkauf, validation in Klattenhoff, C. et al. Drosophila rasiRNA Pathway Mutations Disrupt Embryonic Axis Specification through Activation of an ATR/Chk2 DNA Damage Response. Dev. Cell 12, 45–55 (2007)) |

# Animals and other research organisms

Policy information about studies involving animals; ARRIVE guidelines recommended for reporting animal research, and Sex and Gender in Research

| | |
|---|---|
| Laboratory animals | UAS transgenes: UAS-Cas9 (BDSC: 67086 and BDSC: 58985), UAS-gRNA x14 (this study), UAS-gRNA x27 (this study), UAS-gRNA x66 (this study), UAS-gRNA x223 (this study), UAS-NotchRNAi (VDRC#GD 27229), and Su(var)3–9 11 kb (generated in32, gift from G. Reuter). <br> Mutants: Su(var)3-92 (BDSC: 6210), Su(var)2055 (BDSC: 6234), traKO (BDSC: 67412), and traF (generated in47). <br> Gal4 drivers: nubbin-Gal4 (BDSC: 67086), eyeless-Gal4 (gift from P. Meier), bam-Gal4 (gift from M. Amoyel), and esg-Gal4NP7397, UAS-GFP, Tub-Gal80TS chromosome (gift from J. de Navascués). <br> Reporters: w118E-25 (BDSC: 84091), w118E-10 (BDSC: 84108), and LacZln(3L)BL1 (BDSC: 57370). <br> Sex chromosomes: YBar (BDSC: 81622), YRFP (BDSC: 78567), YBar+yellow (BDSC: 3707), Y21 (this study), Y26 (this study), Y53 (this study), Y69 (this study), Y72 (this study), C(1)DX (BDSC: 64), C(1)M4 (BDSC: 1999), C(1)RM (BDSC: 4248) and C(1,Y)1 (BDSC: 4248). |
| Wild animals | No wild animals were used in the study. |
| Reporting on sex | Both sexes were included. Sex-based analyses were performed. |
| Field-collected samples | No field collected samples were used in the study. |
| Ethics oversight | *Identify the organization(s) that approved or provided guidance on the study protocol, OR state that no ethical approval or guidance was required and explain why not.* |

Note that full information on the approval of the study protocol must also be provided in the manuscript.

