## [Peer Review File · Nature Ecology & Evolution]

Peer Review Information

Journal: Nature Ecology & Evolution

Manuscript Title: Y chromosome toxicity does not contribute to sex-specific differences in longevity

Corresponding author name(s): Renald Delanoue, Bruno Hudry

Editorial Notes:

Reviewer Comments & Decisions:

Decision Letter, initial version:

23rd November 2022

Dear Bruno,

Your manuscript entitled "Y chromosome toxicity does not contribute to sex-specific differences in longevity" has now been seen by three reviewers, whose comments are attached. The reviewers have raised a number of concerns which will need to be addressed before we can offer publication in Nature Ecology & Evolution. We will therefore need to see your responses to the criticisms raised and to some editorial concerns, along with a revised manuscript, before we can reach a final decision regarding publication.

We therefore invite you to revise your manuscript taking into account all reviewer and editor comments. Please highlight all changes in the manuscript text file in Microsoft Word format.

We would like to stress the need to thoroughly discuss previous studies that come to different conclusions as indicated by reviewers 1 and 3.

* If you have not done so already please begin to revise your manuscript so that it conforms to our Article format instructions at <http://www.nature.com/natecolevol/info/final-submission>. Refer also to any guidelines provided in this letter.

2Please use the link below to submit your revised manuscript and related files:

[REDACTED]

Nature Ecology & Evolution is committed to improving transparency in authorship. As part of our efforts in this direction, we are now requesting that all authors identified as 'corresponding author' on published papers create and link their Open Researcher and Contributor Identifier (ORCID) with their account on the Manuscript Tracking System (MTS), prior to acceptance. ORCID helps the scientific community achieve unambiguous attribution of all scholarly contributions. You can create and link your ORCID from the home page of the MTS by clicking on 'Modify my Springer Nature account'. For more information please visit www.springernature.com/orcid.

[REDACTED]

Reviewer expertise:

Reviewer #1: Drosophila evo-devo

Reviewer #2: Chromosome structure, gene regulation, Drosophila genetics

Reviewer #3: Genetics and evolution of Drosophila sex chromosomes

Reviewers' comments:

Reviewer #1 (Remarks to the Author):

The authors generated series of Y chromosome truncations to manipulate the dosage of Y-derived

2heterochromatin, and tested whether heterochromatin dose impacts various aspects of female-male differences. Using these Y deletion series, the authors demonstrate that the heterochromatin dose does not influence male's survival, male-female difference in intestinal physiology (stem cell proliferation, tumor formation caused by Notch-RNAi) and adult body weight. Most critically, they used masculinized XX and feminized XY flies (by manipulating tra) to show that life span correlates with phenotypic sex but not the sex chromosome dosage. Based on these findings, they conclude that they refute the 'toxic Y hypothesis'.

Overall, experiments are well controlled and the results seem to be solid. However, more careful discussion is warranted. By using their own Y chromosome deletion series, they strongly conclude that it is not Y heterochromatin dosage but phenotypic sex that matters. However, the very papers (mainly those from Bachtrog lab) this paper tries to refute are in direct conflict with this paper's conclusion. In earlier work from the Bachtrog lab, they carefully compared Y chromosome doses, and XXY females live shorter than XO males. If the authors want to make the point that Y dose does not matter but phenotypic sex matters, they should consider not only their own data but also conflicting data from previous publications, and provide possible explanations. Without such integration of all data and discussion, the paper will simply leave the readers and the field in confusion. Here I am agnostic as to who (the authors vs. other previous work that supported toxic Y hypothesis) is correct. Indeed, earlier work lacked thorough control of genetic background (which is impossible to achieve, to be fair), so it is quite possible that not the sex chromosome karyotype but something else was causing observed life span differences.

Generally speaking, the work that contradicts previous work should have not only the data to support their own conclusion, but also additional data to explain why the previous work got it wrong (or at least thoroughly discuss why). This is not to unfairly demand a higher standard to the authors of later studies, but it is necessary for progression of science. Without clear discussions as to why previous work might have gotten it wrong, the authors of the previous studies could also easily come back saying 'our data supports our own earlier conclusion' without providing why THIS paper is wrong, leading to a futile cycle. Therefore, I think it is critically important for the authors of this paper to be 'professionally/respectfully confrontational' by providing data or at least discussion why their data and earlier data contradicts.

One critical issue here is that the previous work used the manipulation of sex chromosome karyotype at the whole chromosome level (XO, XXY, XYY etc), whereas this paper used Y chromosome deletion. So, the difference in Y chromosome dose in previous and current work might have an unexpected, non-linear impact. Again, I am not saying which side is right/wrong, but any authors who are trying to refute previous work should attempt to provide some explanations why the authors think the previous study is not correct.

Reviewer #2 (Remarks to the Author):

This is a review of manuscript #NATECOLEVOL-221017731-T, entitled "Y chromosome toxicity does not contribute to sex-specific differences in longevity," and submitted by Renald Delanoue and

3colleagues.

Overall, I find the work to be interesting, and addresses an important problem in chromatin biology, specifically whether heterochromatin in general is a cell-toxic component. I think many of the ideas here are presented elsewhere, but not as decisively, clearly, or comprehensively, and so this work serves a tremendous purpose in addressing a specific question with some finality. Specifically, the idea that heterochromatin contributes to cell or organismal phenotypes linked to age, fecundity, etc – one that I think bears little serious support – is addressed here. The work is clever, simple, convincing, and deserves publication for its importance.

A few comments:

1 - the use of CRISPR to create chromosome terminal deficiencies is interesting by itself. One would not expect to see chromosomes created without telomeric sequence, though of course others have used X-rays (e.g., Mason, Biessmann, Karpen) or induced bridges (e.g., Golic) to create exactly these sorts of structures. This alone is interesting, as it may make a higher-efficiency and sequence-directed means to investigate telomere capping.

2 - I was surprised that standard mitotic (neuroblast) spreads were not used to confirm the structures of the chromosomes as (i) clearly smaller and (ii) lacking the regions expected. The use of flow cytometry is a bit of an oblique – and less-powerful – approach. In my opinion, the paper would be MUCH improved by a DAPI stained image of each chromosome derivative. That could be used to determine whether the short Ys are truly terminal deficiencies, as presented, or are translocations, internal deficiencies, etc. Frankly, this comes across as a pretty glaring oversight given the otherwise-excellent analysis of Y structures in Figure 1.

3 - The analysis of Y effects on PEV is overstated, and should be tempered a bit. The “Mass Action” effect is not uncontroversial, and important papers delving into the biology (by, e.g., Henikoff, Eissenberg, Tartof) are not included. Even work looking at Y dose as suppressor by Janice Spofford has already addressed the issue. Here, it is critical to demonstrate the same, so I do not object to the authors’ experiments or analysis. However, a dose-dependency between Y-linked heterochromatin and degree of suppression is not adequate to distinguish between Mass Action and alternatives. I am merely asking that the data be described, but less conclusion on Mass Action as a mechanism. We and others (Spofford, Maggert, Li, Hartl) have shown that some loci on the Y (here, the rDNA) have disproportionate effects.

4 - Similarly, the authors cannot assert that “epistasis” between HP1 or Su(var)3-9 and heterochromatin proves much at all. Mutation in 205 and 3-9 have the same phenotype as extra heterochromatin, and hyperploidy of 205+ and 3-9+ have the same phenotypes as reduced heterochromatin. The authors cannot disentangle dependent effects (which would indicate epistasis and a common mechanism) from additive effects (which would indicate separate contributions). Again, others’ work (Phalke and Reuter) bears on this, as does ours (Bughio and Maggert), indicting complex interactions between these alleles, each other, and heterochromatin.

It’s not that I don’t believe the data (I do), or that the Y acts as a “sink” for heterochromatin

4components (it does, though calling it a “sink” is a bit silly), but those conclusions are not really supported by these experiments. This issue – the “homogeneity” of heterochromatin is not a settled debate, in fact those of us in the field argue fervently about this (still). More circumspection is called for here, because this issue is not an essential part of this work.

5 - I admit that the ISC/Notch experiments come out of the blue. Perhaps a better explanation is in order. Why is a genetic interaction between Y polymorphism and Notch signaling expected? Beyond that, the experiments are clear and incisive.

6 - The experiments in Supplemental Figure 3 are terrific. Kudos.

7 - The proper genotypes of strains should be included in the Materials and Methods. Stock numbers change! Refer to alleles and background mutations.

Overall, really beautiful work. I will refer to this work in my own, and will teach this work in class. It will also be of interest to others in the field of PEV and heterochromatin, and the importance to those who study aging, genome evolution, and ecology is obvious. I think this warrants publication posthaste.

-Keith A. Maggert

Reviewer #3 (Remarks to the Author):

Delanou and colleagues generate a library of Y chromosome truncations that they use, together with previously established Y chromosome aneuploidy methods, to test effects of the Y chromosome on the physiology of somatic tissues. They demonstrate that, as previously established, the amount of Y chromosome correlates with position effect variegation in the eye. They also demonstrate that, as previously established, variegation is sensitive to levels of heterochromatin regulators. Both observations are consistent with the idea that the Y chromosome acts as a sink for heterochromatin regulators.

They test several sexually dimorphic physiological phenotypes – ISC proliferation, body size/weight, hyperplastic tumor frequency, and starvation resistance – and find no effect of Y chromosome size variation on these phenotypes. The main surprise is that they are unable to replicate results from Brown et al 2020 suggesting that sex-specific aging depends on the Y chromosome. In contrast, they find that modifications of the sex-determination pathway, independent of the sex chromosomes, do affect aging.

Overall they address an important and interesting topic – nonautonomous epigenetic effects of the Y chromosome. However, in my opinion the current evidence presented does not match the definitive conclusions in the title and abstract. And contradictory studies are not sufficiently discussed to promote healthy scientific discourse.

Specific comments and suggestions:

- Males lacking a Y chromosome in *Drosophila* have been known since Bridges 1916 to be surprisingly normal and healthy. Lack of identified phenotypes associated with the Y is not a surprising conclusion

5in and of itself. It's not clear to me why these specific sexually dimorphic phenotypes were tested or why the authors might have believed that the results would be different from what they found (other than the longevity phenotype)?

- The library of different sized Y chromosomes is interesting and clever. A strength of this method is that comparisons can be made while maintaining a controlled genetic background. The aneuploidy methods they use do not have this advantage. I love that they list the specific genotypes used in each panel but they appear to discount genetic background and the crosses necessary to generate each genotype (generating X0 flies requires a cross). As a result, they appear to be comparing phenotypes of flies derived from different genetic backgrounds – which could lead to misinterpretation of results. For example, in Figure 4A they compare a full length BL78567 Y chromosome to truncations of a Canton S Y chromosome, and it is not clear how they derived the X0 flies. I suggest the authors include any crosses they used to generate each genotype and specifically indicate where different genetic backgrounds are compared.

- They nicely employ different methods to characterize the Y chromosomes they generated - PCR survey, flow cytometry, FISH and phenotyping. But all of these characterization methods are relatively crude – i.e. presence/absence of specific sites on the Y or very rough overall genome size estimates. I am not convinced that the CRISPR-induced modifications actually occur as they anticipate and map out in Figure 1. The figure and writing implies precision editing. How reproducibly did each guide RNA produce fragments of the anticipated size (e.g. flow cytometry of different isogenic lines established after CRISPR with the same guide)? Which guides/guide combinations were used to produce each of the five members of the library (this is never explicitly stated)? Because the Y chromosome is so repetitive and poorly characterized (<50% of sequence known), it would not be surprising if the repeats they target are not contained within the region they expect. I am concerned that they haven't sufficiently addressed whether their Y library contains rearrangements in autosomal or X chromosome regions that may be linked to the Y (so not necessarily removable by outcrossing).

- They say in their abstract that they demonstrate effects on gene silencing “proportional to the level of Y heterochromatin”. The main evidence they present is the PEV phenotypes of their Y size library in Figure 2. While there is clearly a positive correlation in their data, saying that the effect is proportional to Y chromosome size and overlaying linear regressions, as they do, seems misleading. The data in Fig 2, especially B, looks more binary or logistic to me than linear. I also do not agree that a proportional effect of Y chromosome size is necessarily a requirement of the sink model. Different repeats/regions of the Y may be more or less potent as “sinks” for heterochromatin regulators. I suggest the authors take more care with their wording.

A related point - different measurements of the same Y truncation line are not equivalent to measurements between the lines and should not be individually considered when running statistics on whether the phenotype of the lines show a relationship to Y chromosome size.

- Their findings that genetic perturbation of tra can affect longevity is very interesting and warrants further investigation. But an effect of the sex determination pathway on longevity does not preclude an effect of the sex chromosomes on longevity. There is no reason to believe that the two would be mutually exclusive. But the authors' assertion that phenotypic sex determines sex-specific longevity is inconsistent with evidence that females in ZW systems have shorter lifespans than males (see Xirocostas et al 2020, <https://doi.org/10.1098/rsbl.2019.0867>).

- There appears to be a discrepancy in the results presented here and those in Brown et al 2020. But the evidence presented here is not identifiably stronger than the evidence in the previous study, and there is no discussion of potential causes of the discrepancy. I suggest the authors discuss

6experimental differences that they think may have influenced the different outcomes between their studies.

*****END*****

Author Rebuttal to Initial commentsResponse to Reviewers

We thank the Reviewers for their time in examining our manuscript and for providing their valuable comments. We have followed all their suggestions and believe that both the new experiments and their outcome have improved our manuscript. We are providing a point-by-point response to their specific points below. Before doing so, we also highlight the key new data included in the revised manuscript:

- 1- As requested by all the three Reviewers and the editor, we now provide a thorough discussion of previous studies (especially the one from Bachtrog lab that come to different conclusions)
- 2- As demanded by Keith Maggert, we performed mitotic neuroblast spreads
- 3- We extended the phenotypic analysis of the truncated Y chromosomes. We evaluated the reproducibility of our approach and strengthened the point that the Y truncations are truly terminal deficiencies, and not translocations or internal deficiencies.

We present these new data in 6 new Supplementary Figure panels (**Suppl. Fig. 1C-F and H-J**) in a significantly revised manuscript. Changes in the manuscript are highlighted in blue.

Reviewer #1 (Remarks to the Author):

Overall, experiments are well controlled and the results seem to be solid. However, more careful discussion is warranted. By using their own Y chromosome deletion series, they strongly conclude that it is not Y heterochromatin dosage but phenotypic sex that matters. However, the very papers (mainly those from Bachtrog lab) this paper tries to refute are in direct conflict with this paper's conclusion. In earlier work from the Bachtrog lab, they carefully compared Y chromosome doses, and XXY females live shorter than XO males. If the authors want to make the point that Y dose does not matter but phenotypic sex matters, they should consider not only their own data but also conflicting data from previous publications, and provide possible explanations. Without such integration of all data and discussion, the paper will simply leave the readers and the field in confusion. Here I am agnostic as to who (the authors vs. other previous work that supported toxic Y hypothesis) is correct. Therefore, I think it is critically important for the authors of this paper to be 'professionally/respectfully confrontational' by providing data or at least discussion why their data and earlier data contradicts.

Indeed, earlier work lacked thorough control of genetic background (which is impossible to achieve, to be fair), so it is quite possible that not the sex chromosome karyotype but something else was causing observed life span differences.

We agree with the Reviewer's comment that a more thorough discussion of previous studies will help improve our manuscript. We are now providing a new extended Discussion section (**please see pages 9-10, lines 286-91 and 312-335**). The following points are comprehensively discussed:

1- As mentioned by the Reviewer, a potential caveat in the Bachtrog study (Brown et al., 2020 Nat Ecol Evol, PMID: 32313175) is the lack of isogenicity of the genetic background. This is acknowledged by the authors themselves in their discussion: "*Note, however, that we cannot formally exclude background effects using these crosses*" and their title: "*The Y chromosome may contribute to sex-specific ageing in Drosophila*" (Brown et al., 2020 Nat Ecol Evol). This prompted us to develop a new approach using CRISPR-Cas9, to generate males with not only the exact same (*Canton-S*) autosomal genetic background, but also identical sex chromosomes. The only genetic variation is the size of the Y chromosome, all Y deletions deriving from the same Y chromosome. We also addressed this problem by performing experiments in three different genetic contexts: Y deletions (Fig. 4A), manipulation of sex chromosome karyotype at the whole chromosome level (XO, XXY, XYY, Fig. 4B), and *HP¹⁰⁵* mutant conditions (Fig. 4C). This diversity of conditions also allowed us to evaluate genomic background effects to some extent. Indeed, we found robust and identical results using these three independent setups: no contribution of the Y in sex differences in longevity.

2- Unlike our study, the Bachtrog paper (Brown et al., 2020 Nat Ecol Evol) did not include biological replicates. For example, we repeated the XY versus XO comparison 12 times (with 150 flies in each replicate). We did not detect Y's impact on longevity in any replicate.

3- Two recent studies also reported, like us, a lack of effect of the Y chromosome on longevity using different approaches. First, the latest work of the Bachtrog team (Nguyen et al., 2022 Genome Res, PMID: 35501131) studied the natural variation of the Y chromosome size in *D. pseudoobscura*. These polymorphic Y chromosomes differ almost twofold in size, ranging from 30 to 60 Mb (this difference is caused by the accumulation of a handful of active transposable elements). Although males with a larger Y chromosome show slightly increased levels of transposable element expression, and more so in old males, these differences are relatively minor and do not result in faster aging. The authors conclude, like us: "We find no significant difference in longevity between males carrying the differently sized Y chromosomes." Secondly, another recent paper performed by Wu et al., 2020 Aging Cell (PMID: 32069521) generated manipulations of sex chromosome karyotype at the whole chromosome level (XO, XXY, XYY) and did lifespan experiments on different diets (amino acids, sugar or cholesterol restrictions). They didn't observe any differences between XY and XO males or between XX and XXY females. Note that these experiments were performed in a different *Dahomey* background. It is important to emphasise that both studies could not control for the genetic background of the sex chromosomes, unlike us. Thus, the lack of effect of the Y chromosome in longevity is very robust and is found in the *Dahomey* background (Wu et al., 2020 Aging Cell), the *CantonS* background (our study, Fig. 4A), the *HP¹⁰⁵* mutant conditions (our study, Fig. 4C), in *D. pseudoobscura* (Nguyen et al., 2022 Genome Res) and using manipulation of sex chromosome karyotype at the whole chromosome level (XO, XXY, XYY) (Wu et al., 2020 Aging Cell and our study, Fig. 4B) or polymorphic Y of different sizes (Nguyen et al., 2022 Genome Res and our study, Fig. 4A). Inversely, the negative impact of the Y chromosome on longevity was never reproduced since the initial Brown et al., 2020 Nat Ecol Evol paper.

4- To conclude, our current study, beside that the Y chromosome has no impact, identifies the molecular driver of sex differences in longevity in *Drosophila melanogaster*: the female sex determinant *transformer*. Genetic manipulations of this factor allow complete bidirectional transformation in longevity in both sexes independently of the chromosomal sex (Fig. 4D and Suppl. Fig. 3F). Interestingly, a paper from Partridge's lab, published last month, also reported an impact of *transformer* on sex differences in longevity in fly but in the context of lifespan extension in response to rapamycin treatment (Regan et al., 2022 Nat Aging, <https://www.nature.com/articles/s43587-022-00308-Z>). In the nematode *Caenorhabditis elegans*, the terminal effector of the sex-determination pathway, Transformer protein 1 (Tra-1), has also been linked to lifespan extension in hermaphrodites compared to males (Hotzi et al., 2018, Aging Cell, PMID: 29493066).

Overall, we believe that our findings, combined with the recent literature, reject the "toxic Y" hypothesis, which postulates that the Y chromosome leads to reduced lifespan in XY individuals.

One critical issue here is that the previous work used the manipulation of sex chromosome karyotype at the whole chromosome level (XO, XXY, XYY etc), whereas this paper used Y chromosome deletion. So, the difference in Y chromosome dose in previous and current work might have an unexpected, non-linear impact.

We also performed experiments using the manipulation of sex chromosome karyotype at the whole chromosome level (XO, XYY males, and XXY females). These data are presented in Fig. 4B. We believe that this point has been addressed by the above response, but just as a summary: the lack of effect of the Y chromosome in longevity is very robust and is found in the *Dahomey* background (Wu et al., 2020 Aging Cell), the *CantonS* background (our study, Fig. 4A), the *HP¹⁰⁵* mutant conditions (our study, Fig. 4C), in *D. pseudoobscura* (Nguyen et al., 2022 Genome Res) and using manipulation of sex chromosome karyotype at the whole chromosome level (XO, XXY, XYY) (Wu et al., 2020 Aging Cell and our study, Fig. 4B) or polymorphic Y of different sizes (Nguyen et al., 2022 Genome Res and our study, Fig. 4A). Inversely, the negative impact of the Y chromosome on longevity was never reproduced since the initial Brown et al., 2020 Nat Ecol Evol paper. Collectively, we strongly think that these findings dismiss the "toxic Y" hypothesis.

Reviewer #2 (Remarks to the Author):

Overall, I find the work to be interesting, and addresses an important problem in chromatin biology, specifically whether heterochromatin in general is a cell-toxic component. I think many of the ideas here are presented elsewhere, but not as decisively, clearly, or comprehensively, and so this work serves a tremendous purpose in addressing a specific question with some finality. Specifically, the idea that heterochromatin contributes to cell or organismal phenotypes linked to age, fecundity, etc – one that I think bears little serious support – is addressed here. The work is clever, simple, convincing, and deserves publication for its importance.

A few comments:

1 - the use of CRISPR to create chromosome terminal deficiencies is interesting by itself. One would not expect to see chromosomes created without telomeric sequence, though of course others have used X-rays (e.g., Mason, Biessmann, Karpen) or induced bridges (e.g., Golic) to create exactly these sorts of structures. This alone is interesting, as it may make a higher-efficiency and sequence-directed means to investigate telomere capping.

We would like to thank the Reviewer for the positive assessment of our work.

2 - I was surprised that standard mitotic (neuroblast) spreads were not used to confirm the structures of the chromosomes as (i) clearly smaller and (ii) lacking the regions expected. The use of flow cytometry is a bit of an oblique – and less-powerful – approach. In my opinion, the paper would be MUCH improved by a DAPI stained image of each chromosome derivative. That could be used to determine whether the short Ys are truly terminal deficiencies, as presented, or are translocations, internal deficiencies, etc. Frankly, this comes across as a pretty glaring oversight given the otherwise-excellent analysis of Y structures in Figure 1.

We agree with the Reviewer and have now included results of mitotic neuroblast spreads, shown in **Suppl. Fig. 1H**. Additionally, we performed genetic experiments to extend the phenotypic analysis of the modified Y chromosomes to strengthen our results. These analyses demonstrate that they are truly terminal deficiencies, and not translocations or internal deficiencies. These experiments are included in 5 new Supplementary Figure panels (**Suppl. Fig. 1C-F and J**), of the revised manuscript, and are described below (**please see pages 4-5, lines 108-132 and lines 152-55**).

1- The probability of generating inter-chromosomal translocations, using our approach, could be high if the DNA double-strand breaks induced, using gRNAs targeting the Y chromosome, happen also on autosomes or on the X chromosome. We performed additional experiments in the soma to evaluate this potential off-target effect of the gRNAs (Figure to Reviewers 1). Interestingly, expressing Cas9 and the gRNAs in the wing cells induced not only male-specific DNA double-strand breaks (p- γ H2Av marker, panel A), but also male-specific caspase activation (revealed using an antibody recognizing an activated version of the Caspase-3-like effector caspase, cleaved Death caspase-1 (cDcp-1), panel A). This caspase activation induced cell death and elimination specifically in males (panel B) when these cells were placed into a competitive context surrounded by wild-type cells (clonal experiments). Consequently, severe developmental defects were observed in adult wings in males but not females (panel C). All these results indicate that our gRNAs have a very low off-target activity on other chromosomes (autosomes/X chromosome), minimizing the chance of producing inter-chromosomal translocations.

Figure to Reviewers 1.

- (A) Wing discs of late third-instar expressing Cas9 and the *gRNA* x223 under the control of *nub-Gal4* driver (expressed in the wing pouch, dashed lines) and stained for the DNA double-strand breaks marker p-yH2Av (green) and the activated version of the caspase-3-like effector caspase, Death caspase-1 (cDcp-1, red). Scale bars, 50 μ m.
- (B) Clones, marked by GFP (green) and expressing Cas9 and the *gRNA* x223 were generated in the wing discs of late third-instar. The histogram shows the quantification of the number of clones per wing pouch per genotype/sex.
- (C) Cuticle preparations of adult wings expressing Cas9 and the *gRNA* x223 under the control of *nub-Gal4*.

2- In addition, we estimated the actual frequency of Y to X/autosome translocations using a genetic marker. First, using three different *gRNAs* (x27, x66, and x223) and replicates, we assessed the frequency of Y deletions, measured by the loss of an RFP transgene located on the Y chromosome. Depending on the *gRNA*, 16% to 42% of the male progeny carried a deleted Y (Suppl. Fig. 1C). To determine the frequency of Y to X/autosome translocations, in the same crosses, we counted the number of female progeny who gain RFP expression (corresponding to the translocation of the RFP transgene from the Y chromosome to another chromosome). The translocation frequency was between 0.0 to 0.5% (Suppl. Fig. 1D). We also carried out a PCR survey of all the Y chromosome coding genes for all the RFP-positive females. Most of these females were XXY females generated by non-disjunction of the sex chromosomes and did not correspond to translocation events. Consequently, in our method, the inter-chromosomal translocation frequency is at least 80 times lower than the probability to generate Y deletions.

3-We directly visualised if the modified Ys carried additional genetic material coming from the X chromosome (corresponding to X to Y translocations). We performed experiments using *gRNA* x223 and three different X chromosomes carrying GFP transgenes located near the centromere, the telomere, or in the middle of the X chromosome. Then, in the male progeny with deleted Y chromosomes (who have lost the Y-linked RFP transgene), we evaluated the gain of GFP expression (corresponding to the translocation of the GFP transgenes from the X chromosome to the Y chromosome). We could never recover any RFP-negative, GFP-positive males (Suppl. Fig. 1E). These new results indicate that X to Y translocation frequency is extremely low with our approach and that deleted Y chromosomes do not carry additional X chromosome material.

4-To assess the frequency of internal versus terminal Y deletions, we selected 90 deleted Ys, from new crosses, to carry out a PCR survey of all the Y chromosome coding genes. We identified 46 terminal deletions, 44 complete Y deletions (X0 males), and 0 internal deletions (Suppl. Fig. 1F). These results indicate that our modified Ys are terminal deletions.

5-Finally, it is important to note that similar results were obtained in human embryonic stem cell lines by Zuccaro et al., in 2020 Cell, PMID: 33125898. They found that a frequent outcome of Cas9-induced double-strand breaks is the generation of terminal deletions or complete loss of the targeted chromosome.

3 - The analysis of Y effects on PEV is overstated, and should be tempered a bit. The “Mass Action” effect is not uncontroversial, and important papers delving into the biology (by, e.g., Henikoff, Eissenberg, Tartof) are not included. Even work looking at Y dose as suppressor by Janice Spofford has already addressed the issue. Here, it is critical to demonstrate the same, so I do not object to the authors’ experiments or analysis. However, a dose-dependency between Y-linked heterochromatin and degree of suppression is not adequate to distinguish between Mass Action and alternatives.

I am merely asking that the data be described, but less conclusion on Mass Action as a mechanism. We and others (Spofford, Maggert, Li, Hartl) have shown that some loci on the Y (here, the rDNA) have disproportionate effects.

4 - Similarly, the authors cannot assert that “epistasis” between HP1 or Su(var)3-9 and heterochromatin proves much at all. Mutation in 205 and 3-9 have the same phenotype as extra heterochromatin, and hyperploidy of 205+ and 3-9+ have the same phenotypes as reduced heterochromatin. The authors cannot disentangle dependent effects (which would indicate epistasis and a common mechanism) from additive effects (which would indicate separate contributions). Again, others’ work (Phalke and Reuter) bears on this, as does ours (Bughio and Maggert), indicating complex interactions between these alleles, each other, and heterochromatin. It’s not that I don’t believe the data (I do), or that the Y acts as a “sink” for heterochromatin components (it does, though calling it a “sink” is a bit silly), but those conclusions are not really supported by these experiments. This issue – the “homogeneity” of heterochromatin is not a settled debate, in fact those of us in the field argue fervently about this (still). More circumspection is called for here, because this issue is not an essential part of this work.

Points 3 and 4:

We agree with the Reviewer and taking these comments into consideration, we have amended the text accordingly (please see pages 6-7, lines 183-86 and 198-201). These experiments are done here only to validate our deleted Y chromosomes by repeating published results.

5 - I admit that the ISC/Notch experiments come out of the blue. Perhaps a better explanation is in order. Why is a genetic interaction between Y polymorphism and Notch signaling expected? Beyond that, the experiments are clear and incisive.

We now provide an explanation of this specific experiment (please see page 7, lines 209-215). Somatic loss of the Y chromosome is the most common known acquired human mutation (Zhou et al., 2016 Nat Genet, PMID: 27064253; Wright et al., 2017 Nat Genet, PMID: 28346444). Its frequency increases with age and is associated with decreased survival time from all causes, including cancers (Jobling et al., 2017 Nat Rev Genet, PMID: 28555659). Yet no causal link has been established. We wanted to use CRISPR/Cas9-mediated targeted Y chromosome elimination/shortening to model Y chromosome loss during a pathological condition: a cancer model. More specifically, we wanted to test if Y loss/truncation could be responsible for sex differences in tumour incidence. Adult-specific interference with *Notch* resulting in tissue overgrowth, reminiscent of gastrointestinal tumours, is the only described *Drosophila* adult tumours that present a sex bias (Hudry et al., 2016 Nature, PMID: 26887495). This context was used for this specific reason.

6 - The experiments in Supplemental Figure 3 are terrific. Kudos. Thanks ;).

7 - The proper genotypes of strains should be included in the Materials and Methods. Stock numbers change! Refer to alleles and background mutations.

Sure, we now mention the specific alleles and genetic background in the revised Methods (please see page 14, lines 458-80).

Overall, really beautiful work. I will refer to this work in my own, and will teach this work in class. It will also be of interest to others in the field of PEV and heterochromatin, and the importance to those who study aging, genome evolution, and ecology is obvious. I think this warrants publication posthaste.

We thank the Reviewer for its positive comments.

-Keith A. Maggert

Reviewer #3 (Remarks to the Author):

Reviewer #3: Genetics and evolution of *Drosophila* sex chromosomes

Specific comments and suggestions:

- Males lacking a Y chromosome in *Drosophila* have been known since Bridges 1916 to be surprisingly normal and healthy. Lack of identified phenotypes associated with the Y is not a surprising conclusion in and of itself. It's not clear to me why these specific sexually dimorphic phenotypes were tested or why the authors might have believed that the results would be different from what they found (other than the longevity phenotype)?

The Reviewer is right, since Bridges in 1916, it is well known that XO males in fly have testes and male secondary characters. However, these individuals are sterile. The only sentence related to the Y chromosome in the 1913/1916 Bridges' papers (focused on non-disjunction) is: "*XO* son is entirely unaltered in somatic appearance, as to sex-linked characters, but he is absolutely sterile. This difference between XO and XY males proves that the Y has some normal function in *Drosophila*". The fact that the Y chromosome does not control the sexual identity of the sex organs does not preclude other functions in somatic tissues. Indeed, elegant papers described the impacts of the Y on sex differences in gene expression in fly adult somatic cells (for example Lemos et al., 2010 Proc Natl Acad Sci USA, PMID: 20798037).

Consequently, we tested body weight and size for several reasons: 1) these characters are sexually dimorphic, 2) the molecular and cellular mechanisms driving these differences are still not completely understood, 3) Y chromosome impact has not been tested/published and 4) these parameters affect lifespan. As longevity was our main interest, with respects to Y chromosome's impact on longevity, we wanted to exclude any secondary consequences due to size/weight differences. Indeed, Tantawy et al., 1960 Am Nat found a positive correlation between body size and longevity in *Drosophila*. McCabe and Partridge., 1997 Evolution (PMID: 28565476) reported that female *D. melanogaster* reared at cold temperatures lived longer than smaller flies reared at higher temperatures. Consistent with these results is the observation of a positive correlation between body size and longevity in wild populations of *Drosophila buzzatii* (Rodrigues et al., 1999 Evolution, PMID: 28565408).

Regarding the ISC/Notch experiments, somatic loss of the Y chromosome is the most common known acquired human mutation (Zhou et al., 2016 Nat Genet, PMID: 27064253; Wright et al., 2017 Nat Genet, PMID: 28346444). Its frequency increases with age and is associated with decreased survival time from all causes, including cancers (Jobling et al., 2017 Nat Rev Genet, PMID: 28555659). Yet no causal link has been established. We wanted to use CRISPR/Cas9-mediated targeted Y chromosome elimination/shortening to model Y chromosome loss during a pathological condition: a cancer model. More specifically, we wanted to test if Y loss/truncation could be responsible for sex differences in tumour incidence. Adult-specific interference with *Notch* resulting in tissue overgrowth, reminiscent of gastrointestinal tumours, is the only described *Drosophila* adult tumours that present a sex bias (Hudry et al., 2016 Nature, PMID: 26887495). This context was used for this specific reason.

We agree with the Reviewer's comment that a justification for the use of these specific readouts was missing in our initial manuscript. We have hopefully clarified the issue (please see page 7, lines 209-15 and 225-29).

- The library of different sized Y chromosomes is interesting and clever. A strength of this method is that comparisons can be made while maintaining a controlled genetic background. The aneuploidy methods they use do not have this advantage. I love that they list the specific genotypes used in each panel but they appear to discount genetic background and the crosses necessary to generate each genotype (generating XO flies requires a cross). As a result, they appear to be comparing phenotypes of flies derived from different genetic backgrounds – which could lead to misinterpretation of results. For example, in Figure 4A they compare a full length BL78567 Y chromosome to truncations of a Canton S Y chromosome, and it is not clear how they derived the XO flies. I suggest the authors include any crosses they used to generate each genotype and specifically indicate where different genetic backgrounds are compared.

We completely agree with the Reviewer, and we modified our manuscript to clarify this point. We are comparing flies derived from identical and isogenised genetic backgrounds. As mentioned by the Reviewer, a limitation of the paper done

by the Bachtrog lab (Brown et al., 2020 Nat Ecol Evol, PMID: 32313175) is the lack of control of the genetic background. This is the reason why we developed this original approach using CRISPR-Cas9, allowing the generation of males with the exact same (*Canton-S*) autosomal genetic background, and identical sex chromosomes. Thus, we carefully took this issue into consideration. We acknowledge that the exact background used in the different experiments was not sufficiently described in our previous version. We now mention the specific alleles and genetic background for each experiment/panel in the revised Methods (please see page 14, lines 458-80) and in the full genotype section (please see pages 30-37).

- They nicely employ different methods to characterize the Y chromosomes they generated - PCR survey, flow cytometry, FISH and phenotyping. But all of these characterization methods are relatively crude – i.e. presence/absence of specific sites on the Y or very rough overall genome size estimates. I am not convinced that the CRISPR-induced modifications actually occur as they anticipate and map out in Figure 1. The figure and writing implies precision editing. How reproducible did each guide RNA produce fragments of the anticipated size (e.g. flow cytometry of different isogenic lines established after CRISPR with the same guide)?

This is an interesting point. To investigate the capacity of our approach to generate each time the same Y deletions we have conducted the following additional experiments (please see pages 4-5, lines 108-132, and lines 152-55):

1-First, using three different gRNAs (x27, x66, and x223) and replicates, we assessed the frequency of Y deletions, measured by the loss of an RFP transgene located on the Y chromosome. Depending on the gRNA, 16% to 42% of the male progeny carried a deleted Y (Suppl. Fig. 1C). The efficacy of each gRNA was relatively stable between the three experimental replicates.

2-To evaluate the frequency of recovery of internal versus terminal Y deletions, we selected 90 deleted Ys, from our new crosses, and carried out a PCR survey of all the Y chromosome coding genes for these 90 chromosomes. We identified 46 terminal deletions, 44 complete Y deletions (X0 males), and 0 internal deletions (Suppl. Fig. 1G). These results indicate that our short Ys are terminal deletions.

3-Using the PCR mapping of the 46 recovered Y terminal deletions, we evaluated the reproducibility of generating the exact same deletion using a specific gRNA. Each specific Y truncation was always recovered in the 3 replicates at roughly the same frequency, indicating the efficacy of our approach (Suppl. Fig. 1J). However, it is also important to notice that additional deletions (always the same) were obtained for each specific gRNAs. These deletions were not predictable based on the known position of the gRNA sequences on the Y chromosome. The Reviewer is entirely right, this fact, indeed, illustrates that we do not have the entire sequence of the Y chromosome and all the outcomes of our gRNAs cannot be anticipated. We apologise for the lack of clarity in the previous version and this is now emphasized in the revised submission. Indeed, in Suppl. Fig. 1B, we used on purpose a Y chromosome with 2 markers, one on the tip of the short arm, and one on the long arm. In combination with gRNA x14, targeting sequences on the short arm, we showed that we obtained mostly truncations of the targeted arm but also at lower frequency long arm deletions (or even double short and long arm truncations). Inversely, using a gRNA targeting specifically the long arm (x27) generated mostly long-arm truncations, but also short-arm deletions to a lesser extent. We have clarified this point in the new version of the manuscript.

4-Finally, it is important to note that similar results were obtained in human embryonic stem cell lines by Zuccaro et al., in 2020 Cell, PMID: 33125898. They found that a frequent outcome of Cas9-induced double-strand breaks is the generation of terminal deletions or complete loss of the targeted chromosome.

Which guides/guide combinations were used to produce each of the five members of the library (this is never explicitly stated)?

This information is now indicated in the method section. Y₂₁ was generated using gRNA x66, Y₂₆ gRNA x27, Y₅₃ gRNA x66, Y₆₉ gRNA x223, and Y₇₂ gRNA x14. These data are also presented in the new Suppl. Fig. 1J, where the probability of obtaining each specific truncation is quantified (please see page 16, lines 517-18).

Because the Y chromosome is so repetitive and poorly characterized (<50% of sequence known), it would not be surprising if the repeats they target are not contained within the region they expect. I am concerned that they haven't sufficiently addressed whether their Y library contains rearrangements in autosomal or X chromosome regions that may be linked to the Y (so not necessarily removable by outcrossing).

We agree with the Reviewer and we have performed new experiments to address it. We carried out additional genetic experiments to extend the phenotypic analysis of our modified Ys and consolidate the point that they are truly terminal deficiencies, and not translocations or internal deficiencies. These experiments have led to the incorporation of 6 new Supplementary Figure panels (Suppl. Fig. 1C-F and H-J), into our revised manuscript, and are described below (please see pages 4-5, lines 108-132, and lines 152-55).

1- We performed mitotic neuroblast spreads, these results are now shown in Suppl. Fig 1H.

2- The probability of generating inter-chromosomal translocations, using our approach, could be high if the DNA double-strand breaks induced, using our gRNAs targeting the Y chromosome, happen also on autosomes or on the X chromosome. We performed supplementary experiments in the soma to evaluate this potential off-target effect of our gRNAs (Figure to Reviewers 1). Interestingly, expressing Cas9 and our gRNAs in the wing cells induced not only male-specific DNA double-strand breaks (p- γ H2Av marker, panel A), but also male-specific caspase activation (revealed using an antibody recognizing an activated version of the Caspase-3-like effector caspase, cleaved Death caspase-1 (cDcp-1), panel A). This caspase activation induced cell death and elimination specifically in males (panel B), when these cells were placed into a competitive context surrounded by wild-type cells (clonal experiments). Consequently, severe developmental defects were observed in adult wings in males but not females (panel C). All these results indicate that our gRNAs have a very low off-target activity on other chromosomes (autosomes/X chromosome), minimizing the chance of producing inter-chromosomal translocations.

Figure to Reviewers 1.

- (A) Wing discs of late third-instar expressing Cas9 and the gRNA x223 under the control of *nub-Gal4* driver (expressed in the wing pouch, dashed lines) and stained for the DNA double-strand breaks marker p- γ H2Av (green) and the activated version of the caspase-3-like effector caspase, Death caspase-1 (cDcp-1, red). Scale bars, 50 μ m.
- (B) Clones, marked by GFP (green) and expressing Cas9 and the gRNA x223 were generated in the wing discs of late third-instar. The histogram shows the quantification of the number of clones per wing pouch per genotype/sex.
- (C) Cuticle preparations of adult wings expressing Cas9 and the gRNA x223 under the control of *nub-Gal4*.

2- In addition, we estimated the actual frequency of Y to X/autosome translocations using a genetic marker. First, using three different gRNAs (x27, x66, and x223) and replicates, we assessed the frequency of Y deletions, measured by the loss of an RFP transgene located on the Y chromosome. Depending on the gRNA, 16% to 42% of the male progeny carried a deleted Y (Suppl. Fig. 1C). To determine the frequency of Y to X/autosome translocations, in the same crosses, we counted the number of female progeny who gain RFP expression (corresponding to the translocation of the RFP transgene from the Y chromosome to another chromosome). Translocation frequency was between 0.0 to 0.5% (Suppl. Fig. 1D). We also carried out a PCR survey of all the Y chromosome coding genes for all the RFP-positive females. Most of these females were XXY females generated by non-disjunction of the sex chromosomes and do not correspond to translocation events.

Consequently, in our method, the inter-chromosomal translocation frequency is at least 80 times lower than the probability to generate Y deletions.

3-We also visualised directly if the modified Ys carried additional genetic material coming from the X chromosome (corresponding to X to Y translocations). We performed experiments using *gRNA x223* and three different X chromosomes carrying GFP transgenes located near the centromere, the telomere, or in the middle of the X chromosome. Then, in the male progeny with deleted Y chromosomes (who have lost the Y-linked RFP transgene), we evaluated the gain of GFP expression (corresponding to the translocation of the GFP transgenes from the X chromosome to the Y chromosome). We could never recover any RFP-negative, GFP-positive males (Suppl. Fig. 1E). These new results indicate that X to Y translocation frequency is extremely low with our approach and that our deleted Ys do not carry additional X chromosome material.

We believe that the above-described experiments clarified and consolidated the point that our short Ys are truly terminal deficiencies, and not translocations or internal deficiencies.

- They say in their abstract that they demonstrate effects on gene silencing “proportional to the level of Y heterochromatin”. The main evidence they present is the PEV phenotypes of their Y size library in Figure 2. While there is clearly a positive correlation in their data, saying that the effect is proportional to Y chromosome size and overlaying linear regressions, as they do, seems misleading. The data in Fig 2, especially B, looks more binary or logistic to me than linear. I also do not agree that a proportional effect of Y chromosome size is necessarily a requirement of the sink model. Different repeats/regions of the Y may be more or less potent as “sinks” for heterochromatin regulators. I suggest the authors take more care with their wording.

We agree and we have amended the text accordingly, taking this comment into consideration (please see pages 6-7, lines 183-86 and 198-201). In fact, these experiments are done here only to validate our deleted Y chromosomes by repeating published results. Regarding the linear regressions, we redid all of them using linear versus logistic regressions. We then selected the regression with the highest coefficient of determination (R^2) for each panel. Consequently, logistic regressions are now presented in 5 panels (Fig. 2A-C, Suppl. Fig. 2, and Suppl. Fig. 3D).

A related point - different measurements of the same Y truncation line are not equivalent to measurements between the lines and should not be individually considered when running statistics on whether the phenotype of the lines show a relationship to Y chromosome size.

We would like to thank the Reviewer for pointing out this oversight. We have performed all the new regressions using only the median for each line.

- Their findings that genetic perturbation of *tra* can affect longevity is very interesting and warrants further investigation. But an effect of the sex determination pathway on longevity does not preclude an effect of the sex chromosomes on longevity. There is no reason to believe that the two would be mutually exclusive. But the authors' assertion that phenotypic sex determines sex-specific longevity is inconsistent with evidence that females in ZW systems have shorter lifespans than males (see Xirocostas et al 2020, <https://doi.org/10.1098/rsbl.2019.0867>).

Yes, absolutely, sex determination pathway effects and sex chromosome effects are not mutually exclusive. Taking into consideration our study and the literature (a detailed discussion of the literature is provided below), it is now clear, in *Drosophila*, that the female sex determinant Transformer controls the sex gap in longevity, the Y chromosome having no impact. Our data are not in contradiction with the Xirocostas paper. Indeed, this paper describes a correlation between sex chromosome complement and longevity, providing no causal link or mechanisms. In their discussion, Xirocostas et al acknowledge that: “A future direction will be to formally test the hypothesis that the difference in lifespan between sexes is proportional to the proportional difference in chromosome length between sexes”. We believe it was the purpose of our

study. The mechanism we describe, sex determinants controlling at the same time phenotypic sex and sex differences in longevity, could be at play in other species and explain the pattern seen in the Xirocostas paper. For example, in the nematode *Caenorhabditis elegans*, the terminal effector of the sex-determination pathway, Transformer protein 1 (Tra-1), has also been linked to lifespan extension in hermaphrodites compared to males (Hotzi et al., 2018, Aging Cell, PMID: 29493066).

- There appears to be a discrepancy in the results presented here and those in Brown et al 2020. But the evidence presented here is not identifiably stronger than the evidence in the previous study, and there is no discussion of potential causes of the discrepancy. I suggest the authors discuss experimental differences that they think may have influenced the different outcomes between their studies.

We agree with the Reviewer's comment that a thorough discussion of previous studies was missing in our initial manuscript. We are now providing a new extended Discussion section (please see pages 9-10, lines 286-91 and 312-335). The following points are comprehensively discussed:

1- As mentioned by the Reviewer, a potential caveat in the Bachtrog study (Brown et al., 2020 Nat Ecol Evol, PMID: 32313175) is the lack of isogenicity of the genetic background. This is acknowledged by the authors themselves in their discussion: "Note, however, that we cannot formally exclude background effects using these crosses" and their title: "The Y chromosome may contribute to sex-specific ageing in *Drosophila*" (Brown et al., 2020 Nat Ecol Evol). This prompted us to develop a new approach using CRISPR-Cas9, to generate males with not only the exact same (*CantonS*) autosomal genetic background, but also identical sex chromosomes. The only genetic variation is the size of the Y chromosome, all Y deletions deriving from the same Y chromosome. We also approached this problem by performing experiments in three different genetic contexts: Y deletions (Fig. 4A), manipulation of sex chromosome karyotype at the whole chromosome level (XO, XXY, XYY, Fig. 4B), and *HP¹⁰⁵* mutant conditions (Fig. 4C). This diversity of conditions also allows us to evaluate genomic background effects to some extent. Indeed, we found robust and identical results using these three independent setups: no contribution of the Y in sex differences in longevity.

2- Unlike our study, the Bachtrog paper (Brown et al., 2020 Nat Ecol Evol) did not include biological replicates. For example, we repeated the XY versus XO comparisons 12 times (with 150 flies in each replicate). We did not detect Y's impact on longevity in any replicate.

3- Two recent studies also reported, like us, a lack of effect of the Y chromosome on longevity using different approaches. First, the latest work of the Bachtrog team (Nguyen et al., 2022 Genome Res, PMID: 35501131) studied the natural variation of the Y chromosome size in *D. pseudoobscura*. These polymorphic Y chromosomes differ almost twofold in size, ranging from 30 to 60 Mb (this difference is caused by the accumulation of a handful of active transposable elements). Although males with a larger Y chromosome show slightly increased levels of transposable element expression, and more so in old males, these differences are relatively minor and do not result in faster aging. The authors conclude, like us: "We find no significant difference in longevity between males carrying the differently sized Y chromosomes." Secondly, another recent paper performed by Wu et al., 2020 Aging Cell (PMID: 32069521) generated manipulations of sex chromosome karyotype at the whole chromosome level (XO, XXY, XYY) and did lifespan experiments on different diets (amino acids, sugar or cholesterol restrictions). They didn't observe any differences between XY and XO males or between XX and XYY females. Note that these experiments were performed in a different *Dahomey* background. It is important to emphasise that both studies could not control for the genetic background of the sex chromosomes, unlike us. Thus, the lack of effect of the Y chromosome in longevity is very robust and is found in the *Dahomey* background (Wu et al., 2020 Aging Cell), the *CantonS* background (our study, Fig. 4A), the *HP¹⁰⁵* mutant conditions (our study, Fig. 4C), in *D. pseudoobscura* (Nguyen et al., 2022 Genome Res) and using manipulation of sex chromosome karyotype at the whole chromosome level (XO, XXY, XYY) (Wu et al., 2020 Aging Cell and our study, Fig. 4B) or polymorphic Y of different sizes (Nguyen et al., 2022 Genome Res and our study, Fig. 4A). Inversely, the negative impact of the Y chromosome on longevity was never reproduced since the initial Brown et al., 2020 Nat Ecol Evol paper.

4- To conclude, not only the Y chromosome has no impact but our current study identifies the molecular driver of sex differences in longevity in *Drosophila melanogaster*: the female sex determinant *transformer*. Genetic manipulations of

this factor allow complete bidirectional transformation in longevity in both sexes independently of the chromosomal sex (Fig. 4D and Suppl. Fig. 3F). Interestingly, a paper from the Partridge lab, published last month, also reported an impact of *transformer* on sex differences in longevity in fly but in the context of lifespan extension in response to rapamycin treatment (Regan et al., 2022 Nat Aging, <https://www.nature.com/articles/s43587-022-00308-7>). In the nematode *Caenorhabditis elegans*, the terminal effector of the sex-determination pathway, Transformer protein 1 (Tra-1), has also been linked to lifespan extension in hermaphrodites compared to males.

Overall, we believe that our findings, combined with the recent literature, dismiss the “toxic Y” hypothesis, which postulates that the Y chromosome leads to reduced lifespan in XY individuals.Decision Letter, first revision:

9th March 2023

Dear Bruno,

Thank you for submitting your revised manuscript "Y chromosome toxicity does not contribute to sex-specific differences in longevity" (NATECOLEVOL-221017731A). It has now been seen again by the original reviewers and their comments are below. The reviewers find that the paper has improved in revision, and therefore we'll be happy in principle to publish it in Nature Ecology & Evolution, pending minor revisions to satisfy the reviewers' final requests and to comply with our editorial and formatting guidelines.

[REDACTED]

Reviewer #1 (Remarks to the Author):

Authors have responded to previous review comments well, and their interpretation (combined with edited text) is now supported well. I think this represents an important step to address the impact of having extra (heterochromatic) chromosome, by using background-controlled, CRISPR generated flies. As a final issue, while I do appreciate the authors' point that this is the only study so far that achieved 'controlled backgrounds'. However, such controlled experiments are limited to their Y chromosome deletion series, but not more drastic karyotype anomalies, which is still in the same situation with previous works? I couldn't find detailed descriptions on how they constructed these flies with extra entire Y chromosome. This means that it is still a possibility that background-control experiments are still limited to 'Y chromosome deletion series' whose impact would be always less than one full Y chromosome. I am not saying that they are wrong, or trying to support Bachtrog conclusions (I really have 'no horse' in the race here), but I think it is important to make it clear what was shown and what can be really concluded, which I believe is most important for the scientific community.

Reviewer #2 (Remarks to the Author):

19This is a re-review of NATECOLEVOL-221017731A, entitled "Y chromosome toxicity does not contribute to sex-specific differences in longevity."

My previous review expressed my excitement for this work, which I stand by. The one experimental request I made of the authors (to provide neuroblast squashes of the derivative Y chromosomes) was accommodated, and I thank them for that. I also appreciate that they took my recommendations to temper their discussions of PEV.

This work stands well and strong, even without trying to explain why others' work is inadequate or reaches erroneous conclusions. The editors/journals should not blanch at publishing this work because other, earlier, work may have reached a different conclusion. That arbitrary favoring of "first" work simply stifles progress, makes scientific progress unnecessarily conservative, and contributes to the replication "crises" we find ourselves in.

-Keith A. Maggert

Reviewer #3 (Remarks to the Author):

The authors addressed my concerns well and in clever ways. I feel that their manuscript describes their observations much more clearly and the arguments are more compelling. I do have a few minor suggestions that I feel would improve the clarity of the manuscript, but I would leave these to the authors' discretion.

- e.g. in Line 89 – I would take a bit more care in how you refer to the target sites of the sgRNAs. For example, my understanding is that the guides target *near* FDY, not the actual FDY locus.
- Supplementary Figure 1A – Pls describe, in the figure legend, what "Ctrl" is. I assume this is with no guide RNAs, but it's not clear. I would also recommend moving this specific image from the supplement to Figure 1. There is high background for this stain, so the side-by-side comparison is necessary to be able to conclude that XX is negative.
- I'm puzzled by the intermediate ("II") Bar phenotype. What do the authors think this is caused by?
- Supplemental Fig 1C – What do the percentages listed in blue above the graph mean? Why does the Y-axis only go to 50% of males?
- The results that are summarized in Supplemental Figure 1F should be presented more fully as a table. All the information you obtained from PCR assaying all the gene locations on the Y chromosome is lost by lumping everything into two categories. Or, if the PCR results really did reveal only two categories – then this should be stated more explicitly and the cutoffs defined.
- The first paragraph of the results section takes up all of page five and goes well into page 6. I would suggest breaking this up into ~4 paragraphs.
- Line 113 – I suggest omitting the word "comprised".
- Line 213 – I find this phrase confusing: "Gut and adult-specific interference with Notch(N)"
- Line 225 – I think you mean "characteristics" rather than "characters"
- Line 279 – I think more careful wording is necessary since you can't possibly test every physiological difference ... "Y chromosome heterochromatin does not cause or notably contribute to three physiological differences between males and females, including the longevity gap which..."

20- Line 319-320 – “unlike us” appears twice in the sentence
- Line 326 – You begin the paragraph with “On the contrary” but it’s not clear what you are referring to – I suggest stating it explicitly e.g. “Contrary to...” Toxic Y? Brown and Bachtrog?
- Lines 466-468 – It’s okay to recognize whom you obtained the Gal4 drivers from, but the original source (i.e. the papers where they were originally published) should be cited.

Our ref: NATECOLEVOL-221017731A

15th March 2023

Dear Dr. Hudry,

Thank you for your patience as we’ve prepared the guidelines for final submission of your Nature Ecology & Evolution manuscript, “Y chromosome toxicity does not contribute to sex-specific differences in longevity” (NATECOLEVOL-221017731A). Please carefully follow the step-by-step instructions provided in the attached file, and add a response in each row of the table to indicate the changes that you have made. Please also check and comment on any additional marked-up edits we have proposed within the text. Ensuring that each point is addressed will help to ensure that your revised manuscript can be swiftly handed over to our production team.

****We would like to start working on your revised paper, with all of the requested files and forms, as soon as possible (preferably within two weeks). Please get in contact with us immediately if you anticipate it taking more than two weeks to submit these revised files.****

In recognition of the time and expertise our reviewers provide to Nature Ecology & Evolution’s editorial process, we would like to formally acknowledge their contribution to the external peer review of your manuscript entitled “Y chromosome toxicity does not contribute to sex-specific differences in longevity”. For those reviewers who give their assent, we will be publishing their names alongside the

21published article.

Nature Ecology & Evolution offers a Transparent Peer Review option for new original research manuscripts submitted after December 1st, 2019. As part of this initiative, we encourage our authors to support increased transparency into the peer review process by agreeing to have the reviewer comments, author rebuttal letters, and editorial decision letters published as a Supplementary item. When you submit your final files please clearly state in your cover letter whether or not you would like to participate in this initiative. Please note that failure to state your preference will result in delays in accepting your manuscript for publication.

Cover suggestions

As you prepare your final files we encourage you to consider whether you have any images or illustrations that may be appropriate for use on the cover of Nature Ecology & Evolution.

Nature Ecology & Evolution has now transitioned to a unified Rights Collection system which will allow our Author Services team to quickly and easily collect the rights and permissions required to publish your work. Approximately 10 days after your paper is formally accepted, you will receive an email in providing you with a link to complete the grant of rights. If your paper is eligible for Open Access, our Author Services team will also be in touch regarding any additional information that may be required to arrange payment for your article.

Please note that *Nature Ecology & Evolution* is a Transformative Journal (TJ). Authors may publish their research with us through the traditional subscription access route or make their paper immediately open access through payment of an article-processing charge (APC). Authors will not be required to make a final decision about access to their article until it has been accepted. [Find out more about Transformative Journals](https://www.springernature.com/gp/open-research/transformative-journals)

Authors may need to take specific actions to achieve [22](https://www.springernature.com/gp/open-research/funding/policy-compliance- compliance with funder and institutional open access mandates. If your research is supported by a funder that requires immediate open access (e.g. according to Plan S principles) then you should select the gold OA route, and we will direct you to the compliant route where possible. For authors selecting the subscription publication route, the journal's standard licensing terms will need to be accepted, including https://www.nature.com/nature-portfolio/editorial-policies/self-archiving-and-license-to-publish. Those licensing terms will supersede any other terms that the author or any third party may assert apply to any version of the manuscript.

For information regarding our different publishing models please see our Transformative Journals page. If you have any questions about costs, Open Access requirements, or our legal forms, please contact ASJournals@springernature.com.

[REDACTED]

[REDACTED]

Reviewer #1:

Remarks to the Author:

Authors have responded to previous review comments well, and their interpretation (combined with edited text) is now supported well. I think this represents an important step to address the impact of having extra (heterochromatic) chromosome, by using background-controlled, CRISPR generated flies. As a final issue, while I do appreciate the authors' point that this is the only study so far that achieved 'controlled backgrounds'. However, such controlled experiments are limited to their Y chromosome deletion series, but not more drastic karyotype anomalies, which is still in the same situation with previous works? I couldn't find detailed descriptions on how they constructed these flies with extra entire Y chromosome. This means that it is still a possibility that background-control experiments are still limited to 'Y chromosome deletion series' whose impact would be always less than one full Y chromosome. I am not saying that they are wrong, or trying to support Bachtrog conclusions (I really have 'no horse' in the race here), but I think it is important to make it clear what was shown and what can be really concluded, which I believe is most important for the scientific community.

23Reviewer #2:

Remarks to the Author:

This is a re-review of NATECOLEVOL-221017731A, entitled "Y chromosome toxicity does not contribute to sex-specific differences in longevity."

My previous review expressed my excitement for this work, which I stand by. The one experimental request I made of the authors (to provide neuroblast squashes of the derivative Y chromosomes) was accommodated, and I thank them for that. I also appreciate that they took my recommendations to temper their discussions of PEV.

This work stands well and strong, even without trying to explain why others' work is inadequate or reaches erroneous conclusions. The editors/journals should not blanch at publishing this work because other, earlier, work may have reached a different conclusion. That arbitrary favoring of "first" work simply stifles progress, makes scientific progress unnecessarily conservative, and contributes to the replication "crises" we find ourselves in.

-Keith A. Maggert

Reviewer #3:

Remarks to the Author:

The authors addressed my concerns well and in clever ways. I feel that their manuscript describes their observations much more clearly and the arguments are more compelling. I do have a few minor suggestions that I feel would improve the clarity of the manuscript, but I would leave these to the authors' discretion.

- e.g. in Line 89 – I would take a bit more care in how you refer to the target sites of the sgRNAs. For example, my understanding is that the guides target *near* FDY, not the actual FDY locus.
- Supplementary Figure 1A – Pls describe, in the figure legend, what "Ctrl" is. I assume this is with no guide RNAs, but it's not clear. I would also recommend moving this specific image from the supplement to Figure 1. There is high background for this stain, so the side-by-side comparison is necessary to be able to conclude that XX is negative.
- I'm puzzled by the intermediate ("II") Bar phenotype. What do the authors think this is caused by?
- Supplemental Fig 1C – What do the percentages listed in blue above the graph mean? Why does the Y-axis only go to 50% of males?
- The results that are summarized in Supplemental Figure 1F should be presented more fully as a table. All the information you obtained from PCR assaying all the gene locations on the Y chromosome is lost by lumping everything into two categories. Or, if the PCR results really did reveal only two categories – then this should be stated more explicitly and the cutoffs defined.
- The first paragraph of the results section takes up all of page five and goes well into page 6. I would suggest breaking this up into ~4 paragraphs.
- Line 113 – I suggest omitting the word "comprised".
- Line 213 – I find this phrase confusing: "Gut and adult-specific interference with Notch(N)"
- Line 225 – I think you mean "characteristics" rather than "characters"

24- Line 279 – I think more careful wording is necessary since you can't possibly test every physiological difference ... "Y chromosome heterochromatin does not cause or notably contribute to three physiological differences between males and females, including the longevity gap which..."
- Line 319-320 – "unlike us" appears twice in the sentence
- Line 326 – You begin the paragraph with "On the contrary" but it's not clear what you are referring to – I suggest stating it explicitly e.g. "Contrary to..." Toxic Y? Brown and Bachtrog?
- Lines 466-468 – It's okay to recognize whom you obtained the Gal4 drivers from, but the original source (i.e. the papers where they were originally published) should be cited.

Author Rebuttal, first revision:

Response to Reviewers

We thank the Reviewers for their time in examining our manuscript and for providing their valuable comments. We are providing a point-by-point response to their specific points below.

Reviewer #1:

As a final issue, while I do appreciate the authors' point that this is the only study so far that achieved 'controlled backgrounds'. However, such controlled experiments are limited to their Y chromosome deletion series, but not more drastic karyotype anomalies, which is still in the same situation with previous works? I couldn't find detailed descriptions on how they constructed these flies with extra entire Y chromosome. This means that it is still a possibility that background-control experiments are still limited to 'Y chromosome deletion series' whose impact would be always less than one full Y chromosome.

Background-control experiments include the Y chromosome deletion series, as well as X0 males (complete Y deletion). We agree with the Reviewer's comment that for the XYY males, the genetic background of the X^Y linked chromosome cannot be completely controlled. For this X^{YY} males, all the autosomes and the free Y chromosome have been backcrossed or controlled, unlike the X^Y chromosome that cannot be backcrossed. The following sentence was added to the manuscript (**please see page 14, line 511**): "For lifespan experiments, all fly stocks (including the Y truncation lines) used in this study were back-crossed for six or more generations into the laboratory's own *Canton-S* genetic stock, except for the *C(1,Y)1* (BL#4248) chromosome that cannot be backcrossed."

Reviewer #3:

Remarks to the Author:

25- e.g. in Line 89 – I would take a bit more care in how you refer to the target sites of the sgRNAs. For example, my understanding is that the guides target *near* FDY, not the actual FDY locus.

Against was replaced by near to answer the reviewer comment (please see page 3, line 89).

- Supplementary Figure 1A – Pls describe, in the figure legend, what “Ctrl” is. I assume this is with no guide RNAs, but it’s not clear.

“Ctrl” was replaced by no gRNA to answer the reviewer comment (please see Extended Data Fig. 1A).

- I’m puzzled by the intermediate (“II”) Bar phenotype. What do the authors think this is caused by?

The relative efficacy of the method is causing the II Bar phenotype. When Cas9 is not cleaving the Y chromosome, phenotype I is induced. When Cas9 cleaves the Y chromosome in only a few cells, an intermediate Bar phenotype (II) is obtained. Finally, when Cas9 cleaves the Y chromosome in all the eye cells, without exception, a complete rescue (phenotype III) is generated. The differential efficacy of cleavage is determined by the specific gRNA used.

- Supplemental Fig 1C – Why does the Y-axis only go to 50% of males?

We would like to thank the Reviewer for pointing out this oversight. In the corrected version the Y-axis goes up to 100%. (please see Extended Data Fig. 1C).

- The results that are summarized in Supplemental Figure 1F should be presented more fully as a table. All the information you obtained from PCR assaying all the gene locations on the Y chromosome is lost by lumping everything into two categories. Or, if the PCR results really did reveal only two categories – then this should be stated more explicitly and the cutoffs defined.

We agree with the Reviewer’s comment the results that are summarized in Extended Data Fig. 1F are now also presented fully in a Table (please see Supplementary Information, Table 1).

- The first paragraph of the results section takes up all of page five and goes well into page 6. I would suggest breaking this up into ~4 paragraphs.

As suggested by the Reviewer the first paragraph was broken into 3 paragraphs.

- Line 113 – I suggest omitting the word “comprised”.

As suggested by the Reviewer the word “comprised” has been removed.

- Line 213 – I find this phrase confusing: “Gut and adult-specific interference with Notch(N)”

To clarify this point, gut has been replaced by intestinal stem cell.

- Line 225 – I think you mean “characteristics” rather than “characters”

Yes, it is corrected in the new version.

- Line 279 – I think more careful wording is necessary since you can’t possibly test every physiological difference ... “Y chromosome heterochromatin does not cause or notably contribute to three physiological differences between males and females, including the longevity gap which...”

To be more careful, as suggested by the Reviewer, the sentence was modified (**please see page 8, lines 266-8**): *“Collectively, our results strongly demonstrate that Y chromosome heterochromatin does not cause, or have a notable contribution to the tested physiological differences between males and females: this includes the longevity gap which has been recorded in a wide spectrum of species.”*

- Line 319-320 – “unlike us” appears twice in the sentence

Yes, it is corrected in the new version.

- Line 326 – You begin the paragraph with “On the contrary” but it’s not clear what you are referring to – I suggest stating it explicitly e.g. “Contrary to...” Toxic Y? Brown and Bachtrog?

As suggested by the Reviewer, the sentence was modified (please see page 9, line 314): “Contrary to the “toxic Y hypothesis”, we discovered that sex-specific differences in longevity...”

- Lines 466-468 – It’s okay to recognize whom you obtained the Gal4 drivers from, but the original source (i.e. the papers where they were originally published) should be cited.

As suggested by the Reviewer, the original papers where these Gal4 drivers were originally published are now cited.

Final Decision Letter:

14th April 2023

Dear Bruno,

We are pleased to inform you that your Article entitled "Y chromosome toxicity does not contribute to sex-specific differences in longevity", has now been accepted for publication in Nature Ecology & Evolution.

Over the next few weeks, your paper will be copyedited to ensure that it conforms to Nature Ecology and Evolution style. Once your paper is typeset, you will receive an email with a link to choose the appropriate publishing options for your paper and our Author Services team will be in touch regarding any additional information that may be required

You will not receive your proofs until the publishing agreement has been received through our system

Due to the importance of these deadlines, we ask you please us know now whether you will be difficult to contact over the next month. If this is the case, we ask you provide us with the contact information (email, phone and fax) of someone who will be able to check the proofs on your behalf, and who will be available to address any last-minute problems . Once your paper has been scheduled for online publication, the Nature press office will be in touch to confirm the details.

Acceptance of your manuscript is conditional on all authors' agreement with our publication policies (see www.nature.com/authors/policies/index.html). In particular your manuscript must not be published elsewhere and there must be no announcement of the work to any media outlet until the publication date (the day on which it is uploaded onto our web site).

Please note that Nature Ecology & Evolution is a Transformative Journal (TJ). Authors may

28publish their research with us through the traditional subscription access route or make their paper immediately open access through payment of an article-processing charge (APC). Authors will not be required to make a final decision about access to their article until it has been accepted. Find out more about Transformative Journals

Authors may need to take specific actions to achieve compliance with funder and institutional open access mandates. If your research is supported by a funder that requires immediate open access (e.g. according to Plan S principles) then you should select the gold OA route, and we will direct you to the compliant route where possible. For authors selecting the subscription publication route, the journal's standard licensing terms will need to be accepted, including https://www.nature.com/nature-portfolio/editorial-policies/self-archiving-and-license-to-publish. Those licensing terms will supersede any other terms that the author or any third party may assert apply to any version of the manuscript.

An online order form for reprints of your paper is available at https://www.nature.com/reprints/author-reprints.html. All co-authors, authors' institutions and authors' funding agencies can order reprints using the form appropriate to their geographical region.

We welcome the submission of potential cover material (including a short caption of around 40 words) related to your manuscript; suggestions should be sent to Nature Ecology & Evolution as electronic files (the image should be 300 dpi at 210 x 297 mm in either TIFF or JPEG format). Please note that such pictures should be selected more for their aesthetic appeal than for their scientific content, and that colour images work better than black and white or grayscale images. Please do not try to design a cover with the Nature Ecology & Evolution logo etc., and please do not submit composites of images related to your work. I am sure you will understand that we cannot make any promise as to whether any of your suggestions might be selected for the cover of the journal.

You can generate the link yourself when you receive your article DOI by entering it here: <http://authors.springernature.com/share>.

[REDACTED]

P.S. Click on the following link if you would like to recommend Nature Ecology & Evolution to your librarian <http://www.nature.com/subscriptions/recommend.html#forms>

** Visit the Springer Nature Editorial and Publishing website at http://editorial-jobs.springernature.com?utm_source=ejp_NEcoE_email&utm_medium=ejp_NEcoE_email&utm_campaign=ejp_NEcoE for more information about our career opportunities. If you have any questions please click [here](mailto:editorial.publishing.jobs@springernature.com). **